# GUIDE ACTOR-CRITIC FOR CONTINUOUS CONTROL

**Voot Tangkaratt**
RIKEN AIP, Tokyo, Japan
voot.tangkaratt@riken.jp

**Abbas Abdolmaleki**
The University of Aveiro, Aveiro, Portugal
abbas.a@ua.pt

**Masashi Sugiyama**
RIKEN AIP, Tokyo, Japan
The University of Tokyo, Tokyo, Japan
masashi.sugiyama@riken.jp

## ABSTRACT

Actor-critic methods solve reinforcement learning problems by updating a parameterized policy known as an actor in a direction that increases an estimate of the expected return known as a critic. However, existing actor-critic methods only use values or gradients of the critic to update the policy parameter. In this paper, we propose a novel actor-critic method called the guide actor-critic (GAC). GAC firstly learns a guide actor that locally maximizes the critic and then it updates the policy parameter based on the guide actor by supervised learning. Our main theoretical contributions are two folds. First, we show that GAC updates the guide actor by performing second-order optimization in the action space where the curvature matrix is based on the Hessians of the critic. Second, we show that the deterministic policy gradient method is a special case of GAC when the Hessians are ignored. Through experiments, we show that our method is a promising reinforcement learning method for continuous controls.

## 1 INTRODUCTION

The goal of reinforcement learning (RL) is to learn an optimal policy that lets an agent achieve the maximum cumulative rewards known as the return (Sutton & Barto, 1998). Reinforcement learning has been shown to be effective in solving challenging artificial intelligence tasks such as playing games (Mnih et al., 2015; Silver et al., 2016) and controlling robots (Deisenroth et al., 2013; Levine et al., 2016).

Reinforcement learning methods can be classified into three categories: *value-based*, *policy-based*, and *actor-critic* methods. Value-based methods learn an optimal policy by firstly learning a *value function* that estimates the expected return. Then, they infer an optimal policy by choosing an action that maximizes the learned value function. Choosing an action in this way requires solving a maximization problem which is not trivial for continuous controls. While extensions to continuous controls were considered recently, they are restrictive since specific structures of the value function are assumed (Gu et al., 2016; Amos et al., 2017).

On the other hand, policy-based methods, also called policy search methods (Deisenroth et al., 2013), learn a parameterized policy maximizing a sample approximation of the expected return without learning the value function. For instance, policy gradient methods such as REIN-FORCE (Williams, 1992) use gradient ascent to update the policy parameter so that the probability of observing high sample returns increases. Compared with value-based methods, policy search methods are simpler and naturally applicable to continuous problems. Moreover, the sample return is an unbiased estimator of the expected return and methods such as policy gradients are guaranteed to converge to a locally optimal policy under standard regularity conditions (Sutton et al., 1999). However, sample returns usually have high variance and this makes such policy search methods converge too slowly.

Actor-critic methods combine the advantages of value-based and policy search methods. In these methods, the parameterized policy is called an actor and the learned value-function is called a critic.

The goal of these methods is to learn an actor that maximizes the critic. Since the critic is a low variance estimator of the expected return, these methods often converge much faster than policy search methods. Prominent examples of these methods are *actor-critic* (Sutton et al., 1999; Konda & Tsitsiklis, 2003), *natural actor-critic* (Peters & Schaal, 2008), *trust-region policy optimization* (Schulman et al., 2015a), and *asynchronous advantage actor-critic* (Mnih et al., 2016). While their approaches to learn the actor are different, they share a common property that they only use the *value* of the critic, i.e., the zero-th order information, and ignore higher-order ones such as gradients and Hessians w.r.t. actions of the critic[1]. To the best of our knowledge, the only actor-critic methods that use gradients of the critic to update the actor are *deterministic policy gradients* (DPG) (Silver et al., 2014) and *stochastic value gradients* (Heess et al., 2015). However, these two methods do not utilize the second-order information of the critic.

In this paper, we argue that the second-order information of the critic is useful and should not be ignored. A motivating example can be seen by comparing gradient ascent to the Newton method: the Newton method which also uses the Hessian converges to a local optimum in a fewer iterations when compared to gradient ascent which only uses the gradient (Nocedal & Wright, 2006). This suggests that the Hessian of the critic can accelerate actor learning which leads to higher data efficiency. However, the computational complexity of second-order methods is at least quadratic in terms of the number of optimization variables. For this reason, applying second-order methods to optimize the parameterized actor directly is prohibitively expensive and impractical for deep reinforcement learning which represents the actor by deep neural networks.

Our contribution in this paper is a novel actor-critic method for continuous controls which we call *guide actor-critic* (GAC). Unlike existing methods, the actor update of GAC utilizes the second-order information of the critic in a computationally efficient manner. This is achieved by separating actor learning into two steps. In the first step, we learn a *non-parameterized Gaussian actor* that locally maximizes the critic under a Kullback-Leibler (KL) divergence constraint. Then, the Gaussian actor is used as a guide for learning a parameterized actor by supervised learning. Our analysis shows that learning the mean of the Gaussian actor is equivalent to performing a second-order update in the action space where the curvature matrix is given by Hessians of the critic and the step-size is controlled by the KL constraint. Furthermore, we establish a connection between GAC and DPG where we show that DPG is a special case of GAC when the Hessians and KL constraint are ignored.

## 2 BACKGROUND

In this section, we firstly give a background of reinforcement learning. Then, we discuss existing second-order methods for policy learning and their issue in deep reinforcement learning.

### 2.1 REINFORCEMENT LEARNING

We consider discrete-time Markov decision processes (MDPs) with continuous state space $\mathcal{S} \subseteq \mathbb{R}^{d_{\mathbf{s}}}$ and continuous action space $\mathcal{A} \subseteq \mathbb{R}^{d_{\mathbf{a}}}$. We denote the state and action at time step $t \in \mathbb{N}$ by $\boldsymbol{s}_t$ and $\boldsymbol{a}_t$, respectively. The initial state $\boldsymbol{s}_1$ is determined by the initial state density $\boldsymbol{s}_1 \sim p(\boldsymbol{s})$. At time step $t$, the agent in state $\boldsymbol{s}_t$ takes an action $\boldsymbol{a}_t$ according to a policy $\boldsymbol{a}_t \sim \pi(\boldsymbol{a}|\boldsymbol{s}_t)$ and obtains a reward $r_t = r(\boldsymbol{s}_t, \boldsymbol{a}_t)$. Then, the next state $\boldsymbol{s}_{t+1}$ is determined by the transition function $\boldsymbol{s}_{t+1} \sim p(\boldsymbol{s}'|\boldsymbol{s}_t, \boldsymbol{a}_t)$. A trajectory $\tau = (\boldsymbol{s}_1, \boldsymbol{a}_1, r_1, \boldsymbol{s}_2, \dots)$ gives us the cumulative rewards or return defined as $\sum_{t=1}^{\infty} \gamma^{t-1} r(\boldsymbol{s}_t, \boldsymbol{a}_t)$, where the discount factor $0 < \gamma < 1$ assigns different weights to rewards given at different time steps. The expected return of $\pi$ after executing an action $\boldsymbol{a}$ in a state $\boldsymbol{s}$ can be expressed through the *action-value function* which is defined as

$$Q^{\pi}(\boldsymbol{s}, \boldsymbol{a}) = \mathbb{E}_{\pi(\boldsymbol{a}_t|\boldsymbol{s}_t)_{t \geqslant 2}, p(\boldsymbol{s}_{t+1}|\boldsymbol{s}_t, \boldsymbol{a}_t)_{t \geqslant 1}} \left[ \sum_{t=1}^{\infty} \gamma^{t-1} r(\boldsymbol{s}_t, \boldsymbol{a}_t) | \boldsymbol{s}_1 = \boldsymbol{s}, \boldsymbol{a}_1 = \boldsymbol{a} \right], \qquad (1)$$

where $\mathbb{E}_p[\cdot]$ denotes the expectation over the density $p$ and the subscript $t \geqslant 1$ indicates that the expectation is taken over the densities at time steps $t \geqslant 1$. We can define the expected return as

$$\mathcal{J}(\pi) = \mathbb{E}_{p(\boldsymbol{s}_1), \pi(\boldsymbol{a}_t|\boldsymbol{s}_t)_{t \geqslant 1}, p(\boldsymbol{s}_{t+1}|\boldsymbol{s}_t, \boldsymbol{a}_t)_{t \geqslant 1}} \left[ \sum_{t=1}^{\infty} \gamma^{t-1} r(\boldsymbol{s}_t, \boldsymbol{a}_t) \right] = \mathbb{E}_{p(\boldsymbol{s}), \pi(\boldsymbol{a}|\boldsymbol{s})} \left[ Q^{\pi}(\boldsymbol{s}, \boldsymbol{a}) \right]. \qquad (2)$$

---

[1] This is different from using the gradient of the critic w.r.t. critic parameters to update the critic itself.

The goal of reinforcement learning is to find an optimal policy that maximizes the expected return.

The policy search approach (Deisenroth et al., 2013) parameterizes $\pi$ by a parameter $\boldsymbol{\theta} \in \mathbb{R}^{d_\theta}$ and finds $\boldsymbol{\theta}^\star$ which maximizes the expected return:

$$\boldsymbol{\theta}^\star = \arg\max_{\boldsymbol{\theta}} \mathbb{E}_{p(\boldsymbol{s}), \pi_{\boldsymbol{\theta}}(\boldsymbol{a}|\boldsymbol{s})} \left[ Q^{\pi_\theta}(\boldsymbol{s}, \boldsymbol{a}) \right]. \tag{3}$$

*Policy gradient* methods such as REINFORCE (Williams, 1992) solve this optimization problem by gradient ascent:

$$\boldsymbol{\theta} \leftarrow \boldsymbol{\theta} + \alpha \mathbb{E}_{p(\boldsymbol{s}), \pi_{\boldsymbol{\theta}}(\boldsymbol{a}|\boldsymbol{s})} \left[ \nabla_{\boldsymbol{\theta}} \log \pi_{\boldsymbol{\theta}}(\boldsymbol{a}|\boldsymbol{s}) Q^{\pi_\theta}(\boldsymbol{s}, \boldsymbol{a}) \right], \tag{4}$$

where $\alpha > 0$ is a step-size. In policy search, the action-value function is commonly estimated by the *sample return*: $Q^{\pi_\theta}(\boldsymbol{s}, \boldsymbol{a}) \approx \frac{1}{N} \sum_{n,t} \gamma^{t-1} r(\boldsymbol{s}_{t,n}, \boldsymbol{a}_{t,n})$ obtained by collecting $N$ trajectories using $\pi_{\boldsymbol{\theta}}$. The sample return is an unbiased estimator of the action-value function. However, it often has high variance which leads to slow convergence.

An alternative approach is to estimate the action-value function by a *critic* denoted by $\widehat{Q}(\boldsymbol{s}, \boldsymbol{a})$ whose parameter is learned such that $\widehat{Q}(\boldsymbol{s}, \boldsymbol{a}) \approx Q^{\pi_\theta}(\boldsymbol{s}, \boldsymbol{a})$. By replacing the action-value function in Eq.(3) with the critic, we obtain the following optimization problem:

$$\boldsymbol{\theta}^\star = \arg\max_{\boldsymbol{\theta}} \mathbb{E}_{p(\boldsymbol{s}), \pi_{\boldsymbol{\theta}}(\boldsymbol{a}|\boldsymbol{s})} \left[ \widehat{Q}(\boldsymbol{s}, \boldsymbol{a}) \right]. \tag{5}$$

The *actor-critic* method (Sutton et al., 1999) solves this optimization problem by gradient ascent:

$$\boldsymbol{\theta} \leftarrow \boldsymbol{\theta} + \alpha \mathbb{E}_{p(\boldsymbol{s}), \pi_{\boldsymbol{\theta}}(\boldsymbol{a}|\boldsymbol{s})} \left[ \nabla_{\boldsymbol{\theta}} \log \pi_{\boldsymbol{\theta}}(\boldsymbol{a}|\boldsymbol{s}) \widehat{Q}(\boldsymbol{s}, \boldsymbol{a}) \right]. \tag{6}$$

The gradient in Eq.(6) often has less variance than that in Eq.(4), which leads to faster convergence[2]. A large class of actor-critic methods is based on this method (Peters & Schaal, 2008; Mnih et al., 2016). As shown in these papers, these methods only use the value of the critic to learn the actor.

The *deterministic policy gradients* (DPG) method (Silver et al., 2014) is an actor-critic method that uses the first-order information of the critic. DPG updates a deterministic actor $\pi_{\boldsymbol{\theta}}(\boldsymbol{s})$ by

$$\boldsymbol{\theta} \leftarrow \boldsymbol{\theta} + \alpha \mathbb{E}_{p(\boldsymbol{s})} \left[ \nabla_{\boldsymbol{\theta}} \pi_{\boldsymbol{\theta}}(\boldsymbol{s}) \nabla_{\boldsymbol{a}} \widehat{Q}(\boldsymbol{s}, \boldsymbol{a})|_{\boldsymbol{a} = \pi_{\boldsymbol{\theta}}(\boldsymbol{s})} \right]. \tag{7}$$

A method related to DPG is the *stochastic value gradients* (SVG) method (Heess et al., 2015) that is able to learn a stochastic policy but it requires learning a model of the transition function.

The actor-critic methods described above only use up to the first-order information of the critic when learning the actor and ignore higher-order ones. Below, we discuss existing approaches that utilize the second-order information by applying second-order optimization methods to solve Eq.(5).

## 2.2 SECOND-ORDER METHODS FOR POLICY LEARNING

The actor-critic methods described above are *first-order methods* which update the optimization variables based on the gradient of the objective function. First-order methods are popular in deep learning thanks to their computational efficiency. However, it is known in machine learning that *second-order methods* often lead to faster learning because they use the curvature information to compute a better update direction, i.e., the steepest ascent direction along the curvature[3].

The main idea of second-order methods is to rotate the gradient by the inverse of a *curvature matrix*. For instance, second-order updates for the actor-critic method in Eq.(6) are in the following form:

$$\boldsymbol{\theta} \leftarrow \boldsymbol{\theta} + \alpha \boldsymbol{G}^{-1} \left\{ \mathbb{E}_{p(\boldsymbol{s}), \pi_{\boldsymbol{\theta}}(\boldsymbol{a}|\boldsymbol{s})} \left[ \nabla_{\boldsymbol{\theta}} \log \pi_{\boldsymbol{\theta}}(\boldsymbol{a}|\boldsymbol{s}) \widehat{Q}(\boldsymbol{s}, \boldsymbol{a}) \right] \right\}, \tag{8}$$

---

[2]This gradient is a biased estimator of the policy gradient in Eq.(4). However, it is unbiased under some regularity conditions such as compatible function conditions (Sutton et al., 1999).

[3]We use the term "second-order methods" in a broad sense here, including quasi-Newton and natural gradient methods which approximate the curvature matrix by the first-order information.

where $\boldsymbol{G} \in \mathbb{R}^{d_{\boldsymbol{\theta}} \times d_{\boldsymbol{\theta}}}$ is a curvature matrix. The behavior of second-order methods depend on the definition of a curvature matrix. The most well-known second-order method is the *Newton method* where its curvature matrix is the Hessian of the objective function w.r.t. the optimization variables:

$$\boldsymbol{G}_{\text{Hessian}} = \mathbb{E}_{p(\boldsymbol{s}), \pi_{\boldsymbol{\theta}}(\boldsymbol{a}|\boldsymbol{s})} \left[ \left( \nabla_{\boldsymbol{\theta}} \log \pi_{\boldsymbol{\theta}}(\boldsymbol{a}|\boldsymbol{s}) \nabla_{\boldsymbol{\theta}} \log \pi_{\boldsymbol{\theta}}(\boldsymbol{a}|\boldsymbol{s})^{\top} + \nabla_{\boldsymbol{\theta}}^2 \log \pi_{\boldsymbol{\theta}}(\boldsymbol{a}|\boldsymbol{s}) \right) \widehat{Q}(\boldsymbol{s}, \boldsymbol{a}) \right]. \quad (9)$$

The *natural gradient method* is another well-known second-order method which uses the *Fisher information matrix* (FIM) as the curvature matrix (Amari, 1998):

$$\boldsymbol{G}_{\text{FIM}} = \mathbb{E}_{p(\boldsymbol{s}), \pi_{\boldsymbol{\theta}}(\boldsymbol{a}|\boldsymbol{s})} \left[ \nabla_{\boldsymbol{\theta}} \log \pi_{\boldsymbol{\theta}}(\boldsymbol{a}|\boldsymbol{s}) \nabla_{\boldsymbol{\theta}} \log \pi_{\boldsymbol{\theta}}(\boldsymbol{a}|\boldsymbol{s})^{\top} \right]. \quad (10)$$

Unlike the Hessian matrix, FIM provides information about changes of the policy measured by an approximated KL divergence: $\mathbb{E}_{p(\boldsymbol{s})} \left[ \text{KL}(\pi_{\boldsymbol{\theta}}(\boldsymbol{a}|\boldsymbol{s}) || \pi_{\boldsymbol{\theta}'}(\boldsymbol{a}|\boldsymbol{s})) \right] \approx (\boldsymbol{\theta} - \boldsymbol{\theta}')^{\top} \boldsymbol{G}_{\text{FIM}} (\boldsymbol{\theta} - \boldsymbol{\theta}')$ (Kakade, 2001). We can see that $\boldsymbol{G}_{\text{Hessian}}$ and $\boldsymbol{G}_{\text{FIM}}$ are very similar but the former also contains the critic and the Hessian of the actor while the latter does not. This suggests that the Hessian provides more information than that in FIM. However, FIM is always positive semi-definite while the Hessian may be indefinite. Please see Furmston et al. (2016) for detailed comparisons between the two curvature matrices in policy search[4]. Nonetheless, actor-critic methods based on natural gradient were shown to be very efficient (Peters & Schaal, 2008; Schulman et al., 2015a).

We are not aware of existing work that considers second-order updates for DPG or SVG. However, their second-order updates can be trivially derived. For example, a Newton update for DPG is

$$\boldsymbol{\theta} \leftarrow \boldsymbol{\theta} + \alpha \boldsymbol{D}^{-1} \left\{ \mathbb{E}_{p(\boldsymbol{s})} \left[ \nabla_{\boldsymbol{\theta}} \pi_{\boldsymbol{\theta}}(\boldsymbol{s}) \nabla_{\boldsymbol{a}} \widehat{Q}(\boldsymbol{s}, \boldsymbol{a})|_{\boldsymbol{a} = \pi_{\boldsymbol{\theta}}(\boldsymbol{s})} \right] \right\}, \quad (11)$$

where the $(i, j)$-th entry of the Hessian matrix $\boldsymbol{D} \in \mathbb{R}^{d_{\boldsymbol{\theta}} \times d_{\boldsymbol{\theta}}}$ is

$$D_{ij} = \mathbb{E}_{p(\boldsymbol{s})} \left[ \frac{\partial \pi_{\boldsymbol{\theta}}(\boldsymbol{s})}{\partial \theta_i}^{\top} \nabla_{\boldsymbol{a}}^2 \widehat{Q}(\boldsymbol{s}, \boldsymbol{a})|_{\boldsymbol{a} = \pi_{\boldsymbol{\theta}}(\boldsymbol{s})} \frac{\partial \pi_{\boldsymbol{\theta}}(\boldsymbol{s})}{\partial \theta_j} + \frac{\partial^2 \pi_{\boldsymbol{\theta}}(\boldsymbol{s})}{\partial \theta_i \partial \theta_j}^{\top} \nabla_{\boldsymbol{a}} \widehat{Q}(\boldsymbol{s}, \boldsymbol{a})|_{\boldsymbol{a} = \pi_{\boldsymbol{\theta}}(\boldsymbol{s})} \right]. \quad (12)$$

Note that $\partial \pi_{\boldsymbol{\theta}}(\boldsymbol{s}) / \partial \boldsymbol{\theta}$ and $\partial^2 \pi_{\boldsymbol{\theta}}(\boldsymbol{s}) / \partial \boldsymbol{\theta} \partial \boldsymbol{\theta}'$ are vectors since $\pi_{\boldsymbol{\theta}}(\boldsymbol{s})$ is a vector-valued function. Interestingly, the Hessian of DPG contains the Hessians of the actor and the critic. In contrast, the Hessian of the actor-critic method contains the Hessian of the actor and the value of the critic.

Second-order methods are appealing in reinforcement learning because they have high data efficiency. However, inverting the curvature matrix (or solving a linear system) requires cubic computational complexity in terms of the number of optimization variables. For this reason, the second-order updates in Eq.(8) and Eq.(11) are impractical in deep reinforcement learning due to a large number of weight parameters in deep neural networks. In such a scenario, an approximation of the curvature matrix is required to reduce the computational burden. For instance, Furmston et al. (2016) proposed to use only diagonal entries of an approximated Hessian matrix. However, this approximation clearly leads to a loss of useful curvature information since the gradient is scaled but not rotated. More recently, Wu et al. (2017) proposed a natural actor-critic method that approximates block-diagonal entries of FIM. However, this approximation corresponds to ignoring useful correlations between weight parameters in different layers of neural networks.

## 3 GUIDE ACTOR-CRITIC

In this section, we propose the guide actor-critic (GAC) method that performs second-order updates without the previously discussed computational issue. Unlike existing methods that directly learn the parameterized actor from the critic, GAC separates the problem of learning the parameterized actor into problems of 1) learning a *guide actor* that locally maximizes the critic, and 2) learning a parameterized actor based on the guide actor. This separation allows us to perform a second-order update for the guide actor where the dimensionality of the curvature matrix is independent of the parameterization of the actor.

We formulate an optimization problem for learning the guide actor in Section 3.1 and present its solution in Section 3.2. Then in Section 3.3 and Section 3.4, we show that the solution corresponds to performing second-order updates. Finally, Section 3.5 presents the learning step for the parameterized actor using supervised learning. The pseudo-code of our method is provided in Appendix B and the source code is available at https://github.com/voot-t/guide-actor-critic.

---

[4] Furmston et al. (2016) proposed an approximate Newton method for policy search. Their policy search method was shown to perform better than methods based on gradient ascent and natural gradient ascent.

## 3.1 OPTIMIZATION PROBLEM FOR GUIDE ACTOR

Our first goal is to learn a guide actor that maximizes the critic. However, greedy maximization should be avoided since the critic is a noisy estimate of the expected return and a greedy actor may change too abruptly across learning iterations. Such a behavior is undesirable in real-world problems, especially in robotics (Deisenroth et al., 2013). Instead, we maximize the critic with additional constraints:

$$
\begin{aligned}
\max_{\tilde{\pi}} \quad & \mathbb{E}_{p^{\beta}(\boldsymbol{s}), \tilde{\pi}(\boldsymbol{a}|\boldsymbol{s})} \left[ \widehat{Q}(\boldsymbol{s}, \boldsymbol{a}) \right], \\
\text{subject to} \quad & \mathbb{E}_{p^{\beta}(\boldsymbol{s})} \left[ \mathrm{KL}(\tilde{\pi}(\boldsymbol{a}|\boldsymbol{s}) || \pi_{\boldsymbol{\theta}}(\boldsymbol{a}|\boldsymbol{s})) \right] \leqslant \epsilon, \\
& \mathbb{E}_{p^{\beta}(\boldsymbol{s})} \left[ \mathrm{H}(\tilde{\pi}(\boldsymbol{a}|\boldsymbol{s})) \right] \geqslant \kappa, \\
& \mathbb{E}_{p^{\beta}(\boldsymbol{s}) \tilde{\pi}(\boldsymbol{a}|\boldsymbol{s})} = 1,
\end{aligned}
\tag{13}
$$

where $\tilde{\pi}(\boldsymbol{a}|\boldsymbol{s})$ is the guide actor to be learned, $\pi_{\boldsymbol{\theta}}(\boldsymbol{a}|\boldsymbol{s})$ is the current parameterized actor that we want to improve upon, and $p^{\beta}(\boldsymbol{s})$ is the state distribution induced by past trajectories. The objective function differs from the one in Eq.(5) in two important aspects. First, we maximize for a policy function $\tilde{\pi}$ and not for the policy parameter. This is more advantageous than optimizing for a policy parameter since the policy function can be obtained in a closed form, as will be shown in the next subsection. Second, the expectation is defined over a state distribution from past trajectories and this gives us off-policy methods with higher data efficiency. The first constraint is the Kullback-Leibler (KL) divergence constraint where $\mathrm{KL}(p(\boldsymbol{x}) || q(\boldsymbol{x})) = \mathbb{E}_{p(\boldsymbol{x})} \left[ \log p(\boldsymbol{x}) - \log q(\boldsymbol{x}) \right]$. The second constraint is the Shannon entropy constraint where $\mathrm{H}(p(\boldsymbol{x})) = -\mathbb{E}_{p(\boldsymbol{x})} \left[ \log p(\boldsymbol{x}) \right]$. The KL constraint is commonly used in reinforcement learning to prevent unstable behavior due to excessively greedy update (Peters & Schaal, 2008; Peters et al., 2010; Levine & Koltun, 2013; Schulman et al., 2015a). The entropy constraint is crucial for maintaining stochastic behavior and preventing premature convergence (Ziebart et al., 2010; Abdolmaleki et al., 2015; Mnih et al., 2016; Haarnoja et al., 2017). The final constraint ensures that the guide actor is a proper probability density. The KL bound $\epsilon > 0$ and the entropy bound $-\infty < \kappa < \infty$ are hyper-parameters which control the exploration-exploitation trade-off of the method. In practice, we fix the value of $\epsilon$ and adaptively reduce the value of $\kappa$ based on the current actor's entropy, as suggested by Abdolmaleki et al. (2015). More details of these tuning parameters are given in Appendix C.

This optimization problem can be solved by the method of Lagrange multipliers. The solution is

$$
\tilde{\pi}(\boldsymbol{a}|\boldsymbol{s}) \propto \pi_{\boldsymbol{\theta}}(\boldsymbol{a}|\boldsymbol{s})^{\frac{\eta^{\star}}{\eta^{\star}+\omega^{\star}}} \exp \left( \frac{\widehat{Q}(\boldsymbol{s}, \boldsymbol{a})}{\eta^{\star}+\omega^{\star}} \right),
\tag{14}
$$

where $\eta^{\star} > 0$ and $\omega^{\star} > 0$ are dual variables corresponding to the KL and entropy constraints, respectively. The dual variable corresponding to the probability density constraint is contained in the normalization term and is determined by $\eta^{\star}$ and $\omega^{\star}$. These dual variables are obtained by minimizing the dual function:

$$
g(\eta, \omega) = \eta\epsilon - \omega\kappa + (\eta + \omega)\mathbb{E}_{p^{\beta}(\boldsymbol{s})} \left[ \log \int \pi_{\boldsymbol{\theta}}(\boldsymbol{a}|\boldsymbol{s})^{\frac{\eta}{\eta+\omega}} \exp \left( \frac{\widehat{Q}(\boldsymbol{s}, \boldsymbol{a})}{\eta + \omega} \right) \mathrm{d}\boldsymbol{a} \right].
\tag{15}
$$

All derivations and proofs are given in Appendix A. The solution in Eq.(14) tells us that the guide actor is obtained by weighting the current actor with $\widehat{Q}(\boldsymbol{s}, \boldsymbol{a})$. If we set $\epsilon \to 0$ then we have $\tilde{\pi} \approx \pi_{\boldsymbol{\theta}}$ and the actor is not updated. On the other hand, if we set $\epsilon \to \infty$ then we have $\tilde{\pi}(\boldsymbol{a}|\boldsymbol{s}) \propto \exp(\widehat{Q}(\boldsymbol{s}, \boldsymbol{a})/\omega^{\star})$, which is a softmax policy where $\omega^{\star}$ is the temperature parameter.

## 3.2 LEARNING GUIDE ACTOR

Computing $\tilde{\pi}(\boldsymbol{a}|\boldsymbol{s})$ and evaluating $g(\eta, \omega)$ are intractable for an arbitrary $\pi_{\boldsymbol{\theta}}(\boldsymbol{a}|\boldsymbol{s})$. We overcome this issue by imposing two assumptions. First, we assume that the actor is the Gaussian distribution:

$$
\pi_{\boldsymbol{\theta}}(\boldsymbol{a}|\boldsymbol{s}) = \mathcal{N}(\boldsymbol{a}|\boldsymbol{\phi}_{\boldsymbol{\theta}}(\boldsymbol{s}), \boldsymbol{\Sigma}_{\boldsymbol{\theta}}(\boldsymbol{s})),
\tag{16}
$$

where the mean $\boldsymbol{\phi}_{\boldsymbol{\theta}}(\boldsymbol{s})$ and covariance $\boldsymbol{\Sigma}_{\boldsymbol{\theta}}(\boldsymbol{s})$ are functions parameterized by a policy parameter $\boldsymbol{\theta}$. Second, we assume that Taylor's approximation of $\widehat{Q}(\boldsymbol{s}, \boldsymbol{a})$ is locally accurate up to the second-order. More concretely, the second-order Taylor's approximation using an arbitrary action $\boldsymbol{a}_0$ is

given by

$$\widehat{Q}(s, a) \approx \widehat{Q}(s, a_0) + (a - a_0)^\top g_0(s) + \frac{1}{2}(a - a_0)^\top H_0(s)(a - a_0) + \mathcal{O}(\|a\|^3), \quad (17)$$

where $g_0(s) = \nabla_a \widehat{Q}(s, a)|_{a=a_0}$ and $H_0(s) = \nabla_a^2 \widehat{Q}(s, a)|_{a=a_0}$ are the gradient and Hessian of the critic w.r.t. $a$ evaluated at $a_0$, respectively. By assuming that the higher order term $\mathcal{O}(\|a\|^3)$ is sufficiently small, we can rewrite Taylor's approximation at $a_0$ as

$$\widehat{Q}_0(s, a) = \frac{1}{2}a^\top H_0(s)a + a^\top \psi_0(s) + \xi_0(s), \quad (18)$$

where $\psi_0(s) = g_0(s) - H_0(s)a_0$ and $\xi_0(s) = \frac{1}{2}a_0^\top H_0(s)a_0 - a_0^\top g_0(s) + \widehat{Q}(s, a_0)$. Note that $H_0(s)$, $\psi_0(s)$, and $\xi_0(s)$ depend on the value of $a_0$ and do not depend on the value of $a$. This dependency is explicitly denoted by the subscript. The choice of $a_0$ will be discussed in Section 3.3.

Substituting the Gaussian distribution and Taylor's approximation into Eq.(14) yields another Gaussian distribution $\tilde{\pi}(a|s) = \mathcal{N}(a|\phi_+(s), \Sigma_+(s))$, where the mean and covariance are given by

$$\phi_+(s) = F^{-1}(s)L(s), \quad \Sigma_+(s) = (\eta^\star + \omega^\star)F^{-1}(s). \quad (19)$$

The matrix $F(s) \in \mathbb{R}^{d_a \times d_a}$ and vector $L(s) \in \mathbb{R}^{d_a}$ are defined as

$$F(s) = \eta^\star \Sigma_\theta^{-1}(s) - H_0(s), \quad L(s) = \eta^\star \Sigma_\theta^{-1}(s)\phi_\theta(s) + \psi_0(s). \quad (20)$$

The dual variables $\eta^\star$ and $\omega^\star$ are obtained by minimizing the following dual function:

$$\widehat{g}(\eta, \omega) = \eta\epsilon - \omega\kappa + (\eta + \omega)\mathbb{E}_{p^\beta(s)}\left[\log\sqrt{\frac{|2\pi(\eta + \omega)F_\eta^{-1}(s)|}{|2\pi\Sigma_\theta(s)|^{\frac{\eta}{\eta+\omega}}}}\right]$$
$$+ \frac{1}{2}\mathbb{E}_{p^\beta(s)}\left[L_\eta(s)^\top F_\eta^{-1}(s)L_\eta(s) - \eta\phi_\theta(s)^\top \Sigma_\theta^{-1}(s)\phi_\theta(s)\right] + \text{const}, \quad (21)$$

where $F_\eta(s)$ and $L_\eta(s)$ are defined similarly to $F(s)$ and $L(s)$ but with $\eta$ instead of $\eta^\star$.

The practical advantage of using the Gaussian distribution and Taylor's approximation is that the guide actor can be obtained in a closed form and the dual function can be evaluated through matrix-vector products. The expectation over $p^\beta(s)$ can be approximated by e.g., samples drawn from a replay buffer (Mnih et al., 2015). We require inverting $F_\eta(s)$ to evaluate the dual function. However, these matrices are computationally cheap to invert when the dimension of actions is not large.

As shown in Eq.(19), the mean and covariance of the guide actor is computed using both the gradient and Hessian of the critic. Yet, these computations do not resemble second-order updates discussed previously in Section 2.2. Below, we show that for a particular choice of $a_0$, the mean computation corresponds to a second-order update that rotates gradients by a curvature matrix.

### 3.3 GUIDE ACTOR LEARNING AS SECOND-ORDER OPTIMIZATION

For now we assume that the critic is an accurate estimator of the true action-value function. In this case, the quality of the guide actor depends on the accuracy of sample approximation in $\widehat{g}(\eta, \omega)$ and the accuracy of Taylor's approximation. To obtain an accurate Taylor's approximation of $\widehat{Q}(s, a)$ using an action $a_0$, the action $a_0$ should be in the vicinity of $a$. However, we did not directly use any individual $a$ to compute the guide actor, but we weight $\pi_\theta(a|s)$ by $\exp(\widehat{Q}(s, a))$ (see Eq.(14)). Thus, to obtain an accurate Taylor's approximation of the critic, the action $a_0$ needs to be similar to actions sampled from $\pi_\theta(a|s)$. Based on this observation, we propose two approaches to perform Taylor's approximation.

**Taylor's approximation around the mean.** In this approach, we perform Taylor's approximation using the mean of $\pi_\theta(a|s)$. More specifically, we use $a_0 = \mathbb{E}_{\pi_\theta(a|s)}[a] = \phi_\theta(s)$ for Eq.(18). In this case, we can show that the mean update in Eq.(19) corresponds to performing a second-order update in the action space to maximize $\widehat{Q}(s, a)$:

$$\phi_+(s) = \phi_\theta(s) + F_{\phi_\theta}^{-1}(s)\nabla_a \widehat{Q}(s, a)|_{a=\phi_\theta(s)}, \quad (22)$$

where $F_{\phi_\theta}(s) = \eta^\star \Sigma_\theta^{-1}(s) - H_{\phi_\theta}(s)$ and $H_{\phi_\theta}(s) = \nabla_a^2 \widehat{Q}(s, a)|_{a=\phi_\theta(s)}$. This equivalence can be shown by substitution and the proof is given in Appendix A.2. This update equation reveals that the guide actor maximizes the critic by taking a step in the action space similarly to the Newton method. However, the main difference lies in the curvature matrix where the Newton method uses Hessians $H_{\phi_\theta}(s)$ but we use a *damped Hessian* $F_{\phi_\theta}(s)$. The damping term $\eta^\star \Sigma_\theta^{-1}(s)$ corresponds to the effect of the KL constraint and can be viewed as a trust-region that controls the step-size. This damping term is particularly important since Taylor's approximation is accurate only locally and we should not take a large step in each update (Nocedal & Wright, 2006).

**Expectation of Taylor's approximations.** Instead of using Taylor's approximation around the mean, we may use an expectation of Taylor's approximation over the distribution. More concretely, we define $\widetilde{Q}(s, a)$ to be an expectation of $\widehat{Q}_0(s, a)$ over $\pi_\theta(a_0|s)$:

$$\widetilde{Q}(s, a) = \frac{1}{2} a^\top \mathbb{E}_{\pi_\theta(a_0|s)} [H_0(s)] a + a^\top \mathbb{E}_{\pi_\theta(a_0|s)} [\psi_0(s)] + \mathbb{E}_{\pi_\theta(a_0|s)} [\xi_0(s)]. \quad (23)$$

Note that $\mathbb{E}_{\pi_\theta(a_0|s)}[H_0(s)] = \mathbb{E}_{\pi_\theta(a_0|s)}[\nabla_a^2 \widehat{Q}(s, a)|_{a=a_0}]$ and the expectation is computed w.r.t. the distribution $\pi_\theta$ of $a_0$. We use this notation to avoid confusion even though $\pi_\theta(a_0|s)$ and $\pi_\theta(a|s)$ are the same distribution. When Eq.(23) is used, the mean update does not directly correspond to any second-order optimization step. However, under an (unrealistic) independence assumption $\mathbb{E}_{\pi_\theta(a_0|s)}[H_0(s)a_0] = \mathbb{E}_{\pi_\theta(a_0|s)}[H_0(s)]\mathbb{E}_{\pi_\theta(a_0|s)}[a_0]$, we can show that the mean update corresponds to the following second-order optimization step:

$$\phi_+(s) = \phi_\theta(s) + \mathbb{E}_{\pi_\theta(a_0|s)} [F_0(s)]^{-1} \mathbb{E}_{\pi_\theta(a_0|s)} \left[\nabla_a \widehat{Q}(s, a)|_{a=a_0}\right], \quad (24)$$

where $\mathbb{E}_{\pi_\theta(a_0|s)}[F_0(s)] = \eta^\star \Sigma_\theta^{-1}(s) - \mathbb{E}_{\pi_\theta(a_0|s)}[H_0(s)]$. Interestingly, the mean is updated by rotating an expected gradient using an expected Hessians. In practice, the expectations can be approximated using sampled actions $\{a_{0,i}\}_{i=1}^S \sim \pi_\theta(a|s)$. We believe that this sampling can be advantageous for avoiding local optima. Note that when the expectation is approximated by a single sample $a_0 \sim \pi_\theta(a|s)$, we obtain the update in Eq.(24) regardless of the independence assumption.

In the remainder, we use $F(s)$ to denote both of $F_{\phi_\theta}(s)$ and $\mathbb{E}_{\pi_\theta(a_0|s)}[F_0(s)]$, and use $H(s)$ to denote both of $H_{\phi_\theta}(s)$ and $\mathbb{E}_{\pi_\theta(a_0|s)}[H_0(s)]$. In the experiments, we use GAC-0 to refer to GAC with Taylor's approximation around the mean, and we use GAC-1 to refer to GAC with Taylor's approximation by a single sample $a_0 \sim \pi_\theta(a|s)$.

### 3.4 GAUSS-NEWTON APPROXIMATION OF HESSIAN

The covariance update in Eq.(19) indicates that $F(s) = \eta^\star \Sigma_\theta^{-1}(s) - H(s)$ needs to be positive definite. The matrix $F(s)$ is guaranteed to be positive definite if the Hessian matrix $H(s)$ is negative semi-definite. However, this is not guaranteed in practice unless $\widehat{Q}(s, a)$ is a concave function in terms of $a$. To overcome this issue, we firstly consider the following identity:

$$H(s) = \nabla_a^2 \widehat{Q}(s, a) = -\nabla_a \widehat{Q}(s, a)\nabla_a \widehat{Q}(s, a)^\top + \nabla_a^2 \exp(\widehat{Q}(s, a)) \exp(-\widehat{Q}(s, a)). \quad (25)$$

The proof is given in Appendix A.3. The first term is always negative semi-definite while the second term is indefinite. Therefore, a negative semi-definite approximation of the Hessian can be obtained as

$$H_0(s) \approx -\left[\nabla_a \widehat{Q}(s, a)\nabla_a \widehat{Q}(s, a)^\top\right]_{a=a_0}. \quad (26)$$

The second term in Eq.(25) is proportional to $\exp(-\widehat{Q}(s, a))$ and it will be small for high values of $\widehat{Q}(s, a)$. This implies that the approximation should gets more accurate as the policy approach a local maxima of $\widehat{Q}(s, a)$. We call this approximation *Gauss-Newton approximation* since it is similar to the Gauss-Newton approximation for the Newton method (Nocedal & Wright, 2006).

### 3.5 LEARNING PARAMETERIZED ACTOR

The second step of GAC is to learn a parameterized actor that well represents the guide actor. Below, we discuss two supervised learning approaches for learning a parameterized actor.

### 3.5.1 Fully-Parameterized Gaussian Policy

Since the guide actor is a Gaussian distribution with a state-dependent mean and covariance, a natural choice for the parameterized actor is again a parameterized Gaussian distribution with a state-dependent mean and covariance: $\pi_{\boldsymbol{\theta}}(\boldsymbol{a}|\boldsymbol{s}) = \mathcal{N}(\boldsymbol{a}|\boldsymbol{\phi}_{\boldsymbol{\theta}}(\boldsymbol{s}), \boldsymbol{\Sigma}_{\boldsymbol{\theta}}(\boldsymbol{s}))$. The parameter $\boldsymbol{\theta}$ can be learned by minimizing the expected KL divergence to the guide actor:

$$\mathcal{L}^{\mathrm{KL}}(\boldsymbol{\theta}) = \mathbb{E}_{p^{\beta}(\boldsymbol{s})} \left[ \mathrm{KL} \left( \pi_{\boldsymbol{\theta}}(\boldsymbol{a}|\boldsymbol{s}) || \tilde{\pi}(\boldsymbol{a}|\boldsymbol{s}) \right) \right]$$

$$= \mathbb{E}_{p^{\beta}(\boldsymbol{s})} \left[ \frac{\mathrm{Tr}(\boldsymbol{F}(\boldsymbol{s})\boldsymbol{\Sigma}_{\boldsymbol{\theta}}(\boldsymbol{s}))}{\eta^{\star} + \omega^{\star}} - \log |\boldsymbol{\Sigma}_{\boldsymbol{\theta}}(\boldsymbol{s})| \right] + \frac{\mathcal{L}^{\mathrm{W}}(\boldsymbol{\theta})}{\eta^{\star} + \omega^{\star}} + \mathrm{const}, \qquad (27)$$

where $\mathcal{L}^{\mathrm{W}}(\boldsymbol{\theta}) = \mathbb{E}_{p^{\beta}(\boldsymbol{s})} \left[ \|\boldsymbol{\phi}_{\boldsymbol{\theta}}(\boldsymbol{s}) - \boldsymbol{\phi}_+(\boldsymbol{s})\|^2_{\boldsymbol{F}(\boldsymbol{s})} \right]$ is the weighted-mean-squared-error (WMSE) which only depends on $\boldsymbol{\theta}$ of the mean function. The $\mathrm{const}$ term does not depend on $\boldsymbol{\theta}$.

Minimizing the KL divergence reveals connections between GAC and deterministic policy gradients (DPG) (Silver et al., 2014). By computing the gradient of the WMSE, it can be shown that

$$\frac{\nabla_{\boldsymbol{\theta}}\mathcal{L}^{\mathrm{W}}(\boldsymbol{\theta})}{2} = -\mathbb{E}_{p^{\beta}(\boldsymbol{s})} \left[ \nabla_{\boldsymbol{\theta}}\boldsymbol{\phi}_{\boldsymbol{\theta}}(\boldsymbol{s})\nabla_{\boldsymbol{a}}\widehat{Q}(\boldsymbol{s},\boldsymbol{a})|_{\boldsymbol{a}=\boldsymbol{\phi}_{\boldsymbol{\theta}}(\boldsymbol{s})} \right] + \mathbb{E}_{p^{\beta}(\boldsymbol{s})} \left[ \nabla_{\boldsymbol{\theta}}\boldsymbol{\phi}_{\boldsymbol{\theta}}(\boldsymbol{s})\nabla_{\boldsymbol{a}}\widehat{Q}(\boldsymbol{s},\boldsymbol{a})|_{\boldsymbol{a}=\boldsymbol{\phi}_+(\boldsymbol{s})} \right]$$

$$+ \eta^{\star}\mathbb{E}_{p^{\beta}(\boldsymbol{s})} \left[ \nabla_{\boldsymbol{\theta}}\boldsymbol{\phi}_{\boldsymbol{\theta}}(\boldsymbol{s})\boldsymbol{\Sigma}^{-1}(\boldsymbol{s})\boldsymbol{H}_{\boldsymbol{\phi}_{\boldsymbol{\theta}}}^{-1}(\boldsymbol{s})\nabla_{\boldsymbol{a}}\widehat{Q}(\boldsymbol{s},\boldsymbol{a})|_{\boldsymbol{a}=\boldsymbol{\phi}_{\boldsymbol{\theta}}(\boldsymbol{s})} \right]$$

$$- \eta^{\star}\mathbb{E}_{p^{\beta}(\boldsymbol{s})} \left[ \nabla_{\boldsymbol{\theta}}\boldsymbol{\phi}_{\boldsymbol{\theta}}(\boldsymbol{s})\boldsymbol{\Sigma}^{-1}(\boldsymbol{s})\boldsymbol{H}_{\boldsymbol{\phi}_{\boldsymbol{\theta}}}^{-1}(\boldsymbol{s})\nabla_{\boldsymbol{a}}\widehat{Q}(\boldsymbol{s},\boldsymbol{a})|_{\boldsymbol{a}=\boldsymbol{\phi}_+(\boldsymbol{s})} \right]. \qquad (28)$$

The proof is given in Appendix A.4. The negative of the first term is precisely equivalent to DPG. Thus, updating the mean parameter by minimizing the KL loss with gradient descent can be regarded as updating the mean parameter with *biased* DPG where the bias terms depend on $\eta^{\star}$. We can verify that $\nabla_{\boldsymbol{a}}\widehat{Q}(\boldsymbol{s},\boldsymbol{a})|_{\boldsymbol{a}=\boldsymbol{\phi}_+(\boldsymbol{s})} = 0$ when $\eta^{\star} = 0$ and this is the case of $\epsilon \to \infty$. Thus, all bias terms vanish when the KL constraint is ignored and the mean update of GAC coincides with DPG. However, unlike DPG which learns a deterministic policy, we can learn both the mean and covariance in GAC.

### 3.5.2 Gaussian Policy with Parameterized Mean

While a state-dependent parameterized covariance function is flexible, we observe that learning performance is sensitive to the initial parameter of the covariance function. For practical purposes, we propose using a parametrized Gaussian distribution with state-independent covariance: $\pi_{\boldsymbol{\theta}}(\boldsymbol{a}|\boldsymbol{s}) = \mathcal{N}(\boldsymbol{a}|\boldsymbol{\phi}_{\boldsymbol{\theta}}(\boldsymbol{s}), \boldsymbol{\Sigma})$. This class of policies subsumes deterministic policies with additive independent Gaussian noise for exploration. To learn $\boldsymbol{\theta}$, we minimize the mean-squared-error (MSE):

$$\mathcal{L}^{\mathrm{M}}(\boldsymbol{\theta}) = \frac{1}{2}\mathbb{E}_{p^{\beta}(\boldsymbol{s})} \left[ \|\boldsymbol{\phi}_{\boldsymbol{\theta}}(\boldsymbol{s}) - \boldsymbol{\phi}_+(\boldsymbol{s})\|^2_2 \right]. \qquad (29)$$

For the covariance, we use the average of the guide covariances: $\boldsymbol{\Sigma} = (\eta^{\star} + \omega^{\star})\mathbb{E}_{p^{\beta}(\boldsymbol{s})} \left[ \boldsymbol{F}^{-1}(\boldsymbol{s}) \right]$. For computational efficiency, we execute a single gradient update in each learning iteration instead of optimizing this loss function until convergence.

Similarly to the above analysis, the gradient of the MSE w.r.t. $\boldsymbol{\theta}$ can be expanded and rewritten into

$$\nabla_{\boldsymbol{\theta}}\mathcal{L}^{\mathrm{M}}(\boldsymbol{\theta}) = \mathbb{E}_{p^{\beta}(\boldsymbol{s})} \left[ \nabla_{\boldsymbol{\theta}}\boldsymbol{\phi}_{\boldsymbol{\theta}}(\boldsymbol{s})\boldsymbol{H}^{-1}(\boldsymbol{s}) \left( \nabla_{\boldsymbol{a}}\widehat{Q}(\boldsymbol{s},\boldsymbol{a})|_{\boldsymbol{a}=\boldsymbol{\phi}_{\boldsymbol{\theta}}(\boldsymbol{s})} - \nabla_{\boldsymbol{a}}\widehat{Q}(\boldsymbol{s},\boldsymbol{a})|_{\boldsymbol{a}=\boldsymbol{\phi}_+(\boldsymbol{s})} \right) \right]. \qquad (30)$$

Again, the mean update of GAC coincides with DPG when we minimize the MSE and set $\eta^{\star} = 0$ and $\boldsymbol{H}(\boldsymbol{s}) = -\boldsymbol{I}$ where $\boldsymbol{I}$ is the identity matrix. We can also substitute these values back into Eq.(22). By doing so, we can interpret DPG as a method that performs first-order optimization in the action space:

$$\boldsymbol{\phi}_+(\boldsymbol{s}) = \boldsymbol{\phi}_{\boldsymbol{\theta}}(\boldsymbol{s}) + \nabla_{\boldsymbol{a}}\widehat{Q}(\boldsymbol{s},\boldsymbol{a})|_{\boldsymbol{a}=\boldsymbol{\phi}_{\boldsymbol{\theta}}(\boldsymbol{s})}, \qquad (31)$$

and then uses the gradient in Eq.(30) to update the policy parameter. This interpretation shows that DPG is a first-order method that only uses the first-order information of the critic for actor learning. Therefore in principle, GAC, which uses the second-order information of the critic, should learn faster than DPG.

### 3.6 POLICY EVALUATION FOR CRITIC

Beside actor learning, the performance of actor-critic methods also depends on the accuracy of the critic. We assume that the critic $\widehat{Q}_{\boldsymbol{\nu}}(\boldsymbol{s}, \boldsymbol{a})$ is represented by neural networks with a parameter $\boldsymbol{\nu}$. We adopt the approach proposed by Lillicrap et al. (2015) with some adjustment to learn $\boldsymbol{\nu}$. More concretely, we use gradient descent to minimize the squared Bellman error with a slowly moving target critic:

$$\boldsymbol{\nu} \leftarrow \boldsymbol{\nu} - \alpha \nabla_{\boldsymbol{\nu}} \mathbb{E}_{p^{\beta}(\boldsymbol{s}), \beta(\boldsymbol{a}|\boldsymbol{s}), p(\boldsymbol{s}'|\boldsymbol{s}, \boldsymbol{a})} \left[ \left( \widehat{Q}_{\boldsymbol{\nu}}(\boldsymbol{s}, \boldsymbol{a}) - y \right)^2 \right], \tag{32}$$

where $\alpha > 0$ is the step-size. The target value $y = r(\boldsymbol{s}, \boldsymbol{a}) + \gamma \mathbb{E}_{\pi(\boldsymbol{a}'|\boldsymbol{s}')} [\widehat{Q}_{\bar{\boldsymbol{\nu}}}(\boldsymbol{s}', \boldsymbol{a}')]$ is computed by the target critic $\widehat{Q}_{\bar{\boldsymbol{\nu}}}(\boldsymbol{s}', \boldsymbol{a}')$ whose parameter $\bar{\boldsymbol{\nu}}$ is updated by $\bar{\boldsymbol{\nu}} \leftarrow \tau \boldsymbol{\nu} + (1 - \tau)\bar{\boldsymbol{\nu}}$ for $0 < \tau < 1$. As suggested by Lillicrap et al. (2015), the target critic improves the learning stability and we set $\tau = 0.001$ in experiments. The expectation for the squared error is approximated using mini-batch samples $\{(\boldsymbol{s}_n, \boldsymbol{a}_n, r_n, \boldsymbol{s}'_n)\}_{n=1}^{N}$ drawn from a replay buffer. The expectation over the current actor $\pi(\boldsymbol{a}'|\boldsymbol{s}')$ is approximated using samples $\{\boldsymbol{a}'_{n,m}\}_{m=1}^{M} \sim \pi_{\boldsymbol{\theta}}(\boldsymbol{a}'|\boldsymbol{s}'_n)$ for each $\boldsymbol{s}'_n$. We do not use a target actor to compute $y$ since the KL upper-bound already constrains the actor update and a target actor will further slow it down. Note that we are not restricted to this evaluation method and more efficient methods such as *Retrace* (Munos et al., 2016) can also be used.

Our method requires computing $\nabla_{\boldsymbol{a}} \widehat{Q}_{\boldsymbol{\nu}}(\boldsymbol{s}, \boldsymbol{a})$ and its outer product for the Gauss-Newton approximation. The computational complexity of the outer product operation is $\mathcal{O}(d_{\mathbf{a}}^2)$ and is inexpensive when compared to the dimension of $\boldsymbol{\nu}$. For a linear-in-parameter model $\widehat{Q}_{\boldsymbol{\nu}}(\boldsymbol{s}, \boldsymbol{a}) = \boldsymbol{\nu}^{\top} \boldsymbol{\mu}(\boldsymbol{s}, \boldsymbol{a})$, the gradient can be efficiently computed for common choices of the basis function $\boldsymbol{\mu}$ such as the Gaussian function. For deep neural network models, the gradient can be computed by the automatic-differentiation (Goodfellow et al., 2016) where its cost depends on the network architecture.

## 4 RELATED WORK

Besides the connections to DPG, our method is also related to existing methods as follows.

A similar optimization problem to Eq.(13) was considered by the *model-free trajectory optimization* (MOTO) method (Akrour et al., 2016). Our method can be viewed as a non-trivial extension of MOTO with two significant novelties. First, MOTO learns a sequence of time-dependent log-linear Gaussian policies $\pi_t(\boldsymbol{a}|\boldsymbol{s}) = \mathcal{N}(\boldsymbol{a}|\boldsymbol{B}_t\boldsymbol{s} + \boldsymbol{b}_t, \boldsymbol{\Sigma}_t)$, while our method learns a log-nonlinear Gaussian policy. Second, MOTO learns a time-dependent critic given by $\widehat{Q}_t(\boldsymbol{s}, \boldsymbol{a}) = \frac{1}{2}\boldsymbol{a}^{\top}\boldsymbol{C}_t\boldsymbol{a} + \boldsymbol{a}^{\top}\boldsymbol{D}_t\boldsymbol{s} + \boldsymbol{a}^{\top}\boldsymbol{c}_t + \xi_t(\boldsymbol{s})$ and performs policy update with these functions. In contrast, our method learns a more complex critic and performs Taylor's approximation in each training step.

Besides MOTO, the optimization problem also resembles that of *trust region policy optimization* (TRPO) (Schulman et al., 2015a). TRPO solves the following optimization problem:

$$\max_{\boldsymbol{\theta}'} \mathbb{E}_{p^{\pi_{\boldsymbol{\theta}}}(\boldsymbol{s}), \pi_{\boldsymbol{\theta}'}(\boldsymbol{a}|\boldsymbol{s})} \left[ \widehat{Q}(\boldsymbol{s}, \boldsymbol{a}) \right] \text{ subject to } \mathbb{E}_{p^{\pi_{\boldsymbol{\theta}}}(\boldsymbol{s})} \left[ \text{KL}(\pi_{\boldsymbol{\theta}}(\boldsymbol{a}|\boldsymbol{s}) || \pi_{\boldsymbol{\theta}'}(\boldsymbol{a}|\boldsymbol{s})) \right] \leqslant \epsilon, \tag{33}$$

where $\widehat{Q}(\boldsymbol{s}, \boldsymbol{a})$ may be replaced by an estimate of the advantage function (Schulman et al., 2015b). There are two major differences between the two problems. First, TRPO optimizes the policy parameter while we optimize the guide actor. Second, TRPO solves the optimization problem by conjugate gradient where the KL divergence is approximated by the Fisher information matrix, while we solve the optimization problem in a closed form with a quadratic approximation of the critic.

Our method is also related to *maximum-entropy RL* (Ziebart et al., 2010; Azar et al., 2012; Haarnoja et al., 2017; Nachum et al., 2017), which maximizes the expected cumulative reward with an additional entropy bonus: $\sum_{t=1}^{\infty} \mathbb{E}_{p^{\pi}(\boldsymbol{s})} [r(\boldsymbol{s}_t, \boldsymbol{a}_t) + \alpha \text{H}(\pi(\boldsymbol{a}_t|\boldsymbol{s}_t))]$, where $\alpha > 0$ is a trade-off parameter. The optimal policy in maximum-entropy RL is the softmax policy given by

$$\pi_{\text{MaxEnt}}^{\star}(\boldsymbol{a}|\boldsymbol{s}) = \exp \left( \frac{Q_{\text{soft}}^{\star}(\boldsymbol{s}, \boldsymbol{a}) - V_{\text{soft}}^{\star}(\boldsymbol{s})}{\alpha} \right) \propto \exp \left( \frac{Q_{\text{soft}}^{\star}(\boldsymbol{s}, \boldsymbol{a})}{\alpha} \right), \tag{34}$$

where $Q_{\text{soft}}^{\star}(\boldsymbol{s}, \boldsymbol{a})$ and $V_{\text{soft}}^{\star}(\boldsymbol{s})$ are the optimal soft action-value and state-value functions, respectively (Haarnoja et al., 2017; Nachum et al., 2017). For a policy $\pi$, these are defined as

$$Q_{\text{soft}}^{\pi}(\boldsymbol{s}, \boldsymbol{a}) = r(\boldsymbol{s}, \boldsymbol{a}) + \gamma \mathbb{E}_{p(\boldsymbol{s}'|\boldsymbol{s}, \boldsymbol{a})} \left[ V_{\text{soft}}^{\pi}(\boldsymbol{s}') \right], \tag{35}$$

$$V_{\text{soft}}^{\pi}(\boldsymbol{s}) = \alpha \log \int \exp \left( \frac{Q_{\text{soft}}^{\pi}(\boldsymbol{s}, \boldsymbol{a})}{\alpha} \right) \mathrm{d}\boldsymbol{a}. \tag{36}$$

The softmax policy and the soft state-value function in maximum-entropy RL closely resemble the guide actor in Eq.(14) when $\eta^{\star} = 0$ and the log-integral term in Eq.(15) when $\eta = 0$, respectively, except for the definition of action-value functions. To learn the optimal policy of maximum-entropy RL, Haarnoja et al. (2017) proposed *soft Q-learning* which uses importance sampling to compute the soft value functions and approximates the intractable policy using a separate policy function. Our method largely differs from soft Q-learning since we use Taylor's approximation to convert the intractable integral into more convenient matrix-vector products.

The idea of firstly learning a non-parameterized policy and then later learning a parameterized policy by supervised learning was considered previously in *guided policy search* (GPS) (Levine & Koltun, 2013). However, GPS learns the guide policy by trajectory optimization methods such as an iterative linear-quadratic Gaussian regulator (Li & Todorov, 2004), which requires a model of the transition function. In contrast, we learn the guide policy via the critic without learning the transition function.

## 5 EXPERIMENTAL RESULTS

We evaluate GAC on the OpenAI gym platform (Brockman et al., 2016) with the Mujoco Physics simulator (Todorov et al., 2012). The actor and critic are neural networks with two hidden layers of 400 and 300 units, as described in Appendix C. We compare GAC-0 and GAC-1 against deep DPG (DDPG) (Lillicrap et al., 2015), Q-learning with a normalized advantage function (Q-NAF) (Gu et al., 2016), and TRPO (Schulman et al., 2015a;b). Figure 1 shows the learning performance on 9 continuous control tasks. Overall, both GAC-0 and GAC-1 perform comparably with existing methods and they clearly outperform the other methods in Half-Cheetah.

The performance of GAC-0 and GAC-1 is comparable on these tasks, except on Humanoid where GAC-1 learns faster. We expect GAC-0 to be more stable and reliable but easier to get stuck at local optima. On the other hand, the randomness introduced by GAC-1 leads to high variance approximation but this could help escape poor local optima. We conjecture GAC-S that uses $S > 1$ samples for the averaged Taylor's approximation should outperform both GAC-0 and GAC-1. While this is computationally expensive, we can use parallel computation to reduce the computation time.

The expected returns of both GAC-0 and GAC-1 have high fluctuations on the Hopper and Walker2D tasks when compared to TRPO as can be seen in Figure 1g and Figure 1h. We observe that they can learn good policies for these tasks in the middle of learning. However, the policies quickly diverge to poor ones and then they are quickly improved to be good policies again. We believe that this happens because the step-size $\boldsymbol{F}^{-1}(\boldsymbol{s}) = \left( \eta^{\star} \boldsymbol{\Sigma}^{-1} - \boldsymbol{H}(\boldsymbol{s}) \right)^{-1}$ of the guide actor in Eq. (22) can be very large near local optima for Gauss-Newton approximation. That is, the gradients near local optima have small magnitude and this makes the approximation $\boldsymbol{H}(\boldsymbol{s}) = \nabla_{\boldsymbol{a}} \widehat{Q}(\boldsymbol{s}, \boldsymbol{a}) \nabla_{\boldsymbol{a}} \widehat{Q}(\boldsymbol{s}, \boldsymbol{a})^{\top}$ small as well. If $\eta^{\star} \boldsymbol{\Sigma}^{-1}$ is also relatively small then the matrix $\boldsymbol{F}^{-1}(\boldsymbol{s})$ can be very large. Thus, under these conditions, GAC may use too large step sizes to compute the guide actor and this results in high fluctuations in performance. We expect that this scenario can be avoided by reducing the KL bound $\epsilon$ or adding a regularization constant to the Gauss-Newton approximation.

Table 1 in Appendix C shows the wall-clock computation time. DDPG is computationally the most efficient method on all tasks. GAC has low computation costs on tasks with low dimensional actions and its cost increases as the dimensionality of action increases. This high computation cost is due to the dual optimization for finding the step-size parameters $\eta$ and $\omega$. We believe that the computation cost of GAC can be significantly reduced by letting $\eta$ and $\omega$ be external tuning parameters.

## 6 CONCLUSION AND FUTURE WORK

Actor-critic methods are appealing for real-world problems due to their good data efficiency and learning speed. However, existing actor-critic methods do not use second-order information of the

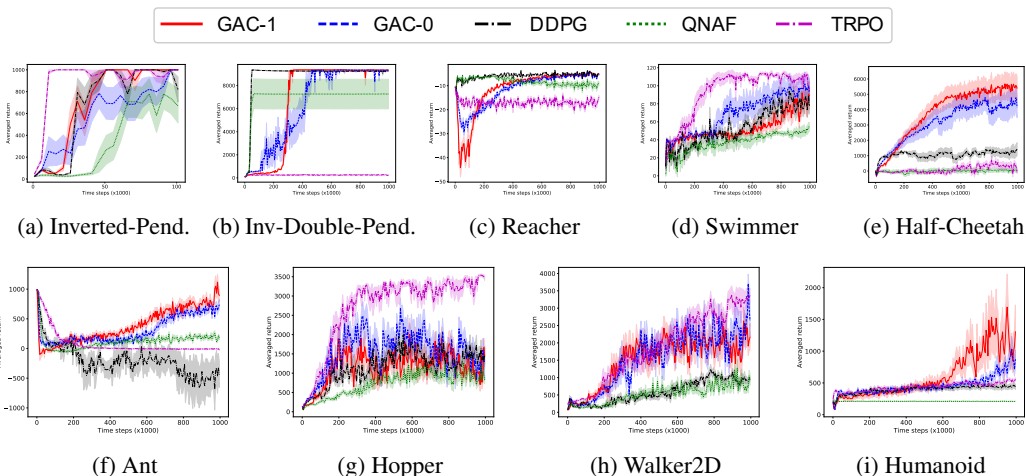

Figure 1: Expected returns averaged over 10 trials. The x-axis indicates training time steps. The y-axis indicates averaged return and higher is better. More clear figures are provided in Appendix C.2.

critic. In this paper, we established a novel framework that distinguishes itself from existing work by utilizing Hessians of the critic for actor learning. Within this framework, we proposed a practical method that uses Gauss-Newton approximation instead of the Hessians. We showed through experiments that our method is promising and thus the framework should be further investigated.

Our analysis showed that the proposed method is closely related to deterministic policy gradients (DPG). However, DPG was also shown to be a limiting case of the stochastic policy gradients when the policy variance approaches zero (Silver et al., 2014). It is currently unknown whether our framework has a connection to the stochastic policy gradients as well, and finding such a connection is our future work.

Our main goal in this paper was to provide a new actor-critic framework and we do not claim that our method achieves the state-of-the-art performance. However, its performance can still be improved in many directions. For instance, we may impose a KL constraint for a parameterized actor to improve its stability, similarly to TRPO (Schulman et al., 2015a). We can also apply more efficient policy evaluation methods such as Retrace (Munos et al., 2016) to achieve better critic learning.

ACKNOWLEDGMENTS

MS was partially supported by KAKENHI 17H00757.

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

# A    DERIVATIONS AND PROOFS

## A.1    DERIVATION OF SOLUTION AND DUAL FUNCTION OF GUIDE ACTOR

The solution of the optimization problem:

$$
\begin{aligned}
\max_{\tilde{\pi}} \quad & \mathbb{E}_{p^{\beta}(\boldsymbol{s}),\tilde{\pi}(\boldsymbol{a}|\boldsymbol{s})}\left[\widehat{Q}(\boldsymbol{s},\boldsymbol{a})\right], \\
\text{subject to} \quad & \mathbb{E}_{p^{\beta}(\boldsymbol{s})}\left[\mathrm{KL}(\tilde{\pi}(\boldsymbol{a}|\boldsymbol{s})||\pi_{\boldsymbol{\theta}}(\boldsymbol{a}|\boldsymbol{s}))\right] \leqslant \epsilon, \\
& \mathbb{E}_{p^{\beta}(\boldsymbol{s})}\left[\mathrm{H}(\tilde{\pi}(\boldsymbol{a}|\boldsymbol{s}))\right] \geqslant \kappa, \\
& \mathbb{E}_{p^{\beta}(\boldsymbol{s})\tilde{\pi}(\boldsymbol{a}|\boldsymbol{s})} = 1,
\end{aligned}
\tag{37}
$$

can be obtained by the method of Lagrange multipliers. The derivation here follows the derivation of similar optimization problems by Peters et al. (2010) and Abdolmaleki et al. (2015). The Lagrangian of this optimization problem is

$$
\begin{aligned}
\mathcal{L}(\tilde{\pi},\eta,\omega,\nu) = {} & \mathbb{E}_{p^{\beta}(\boldsymbol{s}),\tilde{\pi}(\boldsymbol{a}|\boldsymbol{s})}\left[\widehat{Q}(\boldsymbol{s},\boldsymbol{a})\right] + \eta(\epsilon - \mathbb{E}_{p^{\beta}(\boldsymbol{s})}\left[\mathrm{KL}(\tilde{\pi}(\boldsymbol{a}|\boldsymbol{s})||\pi_{\boldsymbol{\theta}}(\boldsymbol{a}|\boldsymbol{s}))\right]) \\
& + \omega(\mathbb{E}_{p^{\beta}(\boldsymbol{s})}\left[\mathrm{H}(\tilde{\pi}(\boldsymbol{a}|\boldsymbol{s}))\right] - \kappa) + \nu(\mathbb{E}_{p^{\beta}(\boldsymbol{s})\tilde{\pi}(\boldsymbol{a}|\boldsymbol{s})} - 1),
\end{aligned}
\tag{38}
$$

where $\eta$, $\omega$, and $\nu$ are the dual variables. Then, by taking derivative of $\mathcal{L}$ w.r.t. $\tilde{\pi}$ we obtain

$$
\partial_{\tilde{\pi}}\mathcal{L} = \mathbb{E}_{p^{\beta}(\boldsymbol{s})}\left[\int\left(\widehat{Q}(\boldsymbol{s},\boldsymbol{a}) - (\eta+\omega)\log\tilde{\pi}(\boldsymbol{a}|\boldsymbol{s}) + \eta\log\pi_{\boldsymbol{\theta}}(\boldsymbol{a}|\boldsymbol{s})\right)\mathrm{d}\boldsymbol{a}\right] - (\eta+\omega-\nu).
\tag{39}
$$

We set this derivation to zero in order to obtain

$$
\begin{aligned}
0 &= \mathbb{E}_{p^{\beta}(\boldsymbol{s})}\left[\int\left(\widehat{Q}(\boldsymbol{s},\boldsymbol{a}) - (\eta+\omega)\log\tilde{\pi}(\boldsymbol{a}|\boldsymbol{s}) + \eta\log\pi_{\boldsymbol{\theta}}(\boldsymbol{a}|\boldsymbol{s})\right)\mathrm{d}\boldsymbol{a}\right] - (\eta+\omega-\nu) \\
&= \widehat{Q}(\boldsymbol{s},\boldsymbol{a}) - (\eta+\omega)\log\tilde{\pi}(\boldsymbol{a}|\boldsymbol{s}) + \eta\log\pi_{\boldsymbol{\theta}}(\boldsymbol{a}|\boldsymbol{s}) - (\eta+\omega-\nu).
\end{aligned}
\tag{40}
$$

Then the solution is given by

$$
\tilde{\pi}(\boldsymbol{a}|\boldsymbol{s}) = \pi_{\boldsymbol{\theta}}(\boldsymbol{a}|\boldsymbol{s})^{\frac{\eta}{\eta+\omega}} \exp\left(\frac{\widehat{Q}(\boldsymbol{s},\boldsymbol{a})}{\eta+\omega}\right)\exp\left(-\frac{\eta+\omega-\nu}{\eta+\omega}\right)
\tag{41}
$$

$$
\propto \pi_{\boldsymbol{\theta}}(\boldsymbol{a}|\boldsymbol{s})^{\frac{\eta}{\eta+\omega}}\exp\left(\frac{\widehat{Q}(\boldsymbol{s},\boldsymbol{a})}{\eta+\omega}\right).
\tag{42}
$$

To obtain the dual function $g(\eta,\omega)$, we substitute the solution to the constraint terms of the Lagrangian and this gives us

$$
\begin{aligned}
\mathcal{L}(\eta,\omega,\nu) = {} & \mathbb{E}_{p^{\beta}(\boldsymbol{s}),\tilde{\pi}(\boldsymbol{a}|\boldsymbol{s})}\left[\widehat{Q}(\boldsymbol{s},\boldsymbol{a})\right] \\
& - (\eta+\omega)\mathbb{E}_{p^{\beta}(\boldsymbol{s}),\tilde{\pi}(\boldsymbol{a}|\boldsymbol{s})}\left[\frac{\widehat{Q}(\boldsymbol{s},\boldsymbol{a})}{\eta+\omega} + \frac{\eta}{\eta+\omega}\log\pi_{\boldsymbol{\theta}}(\boldsymbol{a}|\boldsymbol{s}) - \frac{\eta+\omega-\nu}{\eta+\omega}\right] \\
& + \eta\mathbb{E}_{p^{\beta}(\boldsymbol{s}),\tilde{\pi}(\boldsymbol{a}|\boldsymbol{s})}\left[\log\pi_{\boldsymbol{\theta}}(\boldsymbol{a}|\boldsymbol{s})\right] + \nu\left(\mathbb{E}_{p^{\beta},\tilde{\pi}(\boldsymbol{a}|\boldsymbol{s})} - 1\right) + \eta\epsilon - \omega\kappa.
\end{aligned}
\tag{43}
$$

After some calculation, we obtain

$$
\begin{aligned}
\mathcal{L}(\eta,\omega,\nu) &= \eta\epsilon - \omega\kappa + \mathbb{E}_{p^{\beta}(\boldsymbol{s})}\left[\eta+\omega-\nu\right] \\
&= \eta\epsilon - \omega\kappa + (\eta+\omega)\mathbb{E}_{p^{\beta}(\boldsymbol{s})}\left[\int\pi_{\boldsymbol{\theta}}(\boldsymbol{a}|\boldsymbol{s})^{\frac{\eta}{\eta+\omega}}\exp\left(\frac{\widehat{Q}(\boldsymbol{s},\boldsymbol{a})}{\eta+\omega}\right)\mathrm{d}\boldsymbol{a}\right] \\
&= g(\eta,\omega),
\end{aligned}
\tag{44}
$$

where in the second line we use the fact that $\exp(-\frac{\eta+\omega-\nu}{\eta+\omega})$ is the normalization term of $\tilde{\pi}(\boldsymbol{a}|\boldsymbol{s})$.

A.2 PROOF OF SECOND-ORDER OPTIMIZATION IN ACTION SPACE

Firstly, we show that GAC performs second-order optimization in the action space when Taylor's approximation is performed with $a_0 = \mathbb{E}_{\pi(a|s)}[a] = \phi_\theta(s)$. Recall that Taylor's approximation with $\phi_\theta$ is given by

$$\widehat{Q}(s, a) = \frac{1}{2}a^\top H_{\phi_\theta}(s)a + a^\top \psi_{\phi_\theta}(s) + \xi_{\phi_\theta}(s), \tag{45}$$

where $\psi_{\phi_\theta}(s) = \nabla_a \widehat{Q}(s, a)|_{a=\phi_\theta(s)} - H_{\phi_\theta}(s)\phi_\theta(s)$. By substituting $\psi_{\phi_\theta}(s)$ into $L(s) = \eta^\star \Sigma_\theta^{-1}(s)\phi_\theta(s) - \psi_\theta(s)$, we obtain

$$\begin{aligned} L(s) &= \eta^\star \Sigma_\theta^{-1}(s)\phi_\theta(s) + \nabla_a \widehat{Q}(s, a)|_{a=\phi_\theta(s)} - H_{\phi_\theta}(s)\phi_\theta(s) \\ &= (\eta^\star \Sigma_\theta^{-1}(s) - H_{\phi_\theta}(s))\phi_\theta(s) + \nabla_a \widehat{Q}(s, a)|_{a=\phi_\theta(s)} \\ &= F(s)\phi_\theta(s) + \nabla_a \widehat{Q}(s, a)|_{a=\phi_\theta(s)}. \end{aligned} \tag{46}$$

Therefore, the mean update is equivalent to

$$\begin{aligned} \phi_+(s) &= F^{-1}(s)L(s) \\ &= \phi_\theta(s) + F^{-1}(s)\nabla_a \widehat{Q}(s, a)|_{a=\phi_\theta(s)}, \end{aligned} \tag{47}$$

which is a second-order optimization step with a curvature matrix $F(s) = \eta^\star \Sigma_\theta^{-1}(s) - H_{\phi_\theta}(s)$.

Similarly, for the case where a set of samples $\{a_0\} \sim \pi_\theta(a_0|s) = \mathcal{N}(a_0|\phi_\theta(s), \Sigma(s))$ is used to compute the averaged Taylor's approximation, we obtain

$$L(s) = \eta^\star \Sigma_\theta^{-1}(s)\phi_\theta(s) + \mathbb{E}_{\pi_\theta(a_0|s)}\left[\nabla_a \widehat{Q}(s, a)|_{a=a_0}\right] - \mathbb{E}_{\pi_\theta(a_0|s)}[H_0(s)a_0(s)]. \tag{48}$$

Then, by assuming that $\mathbb{E}_{\pi_\theta(a_0|s)}[H_0(s)a_0(s)] = \mathbb{E}_{\pi_\theta(a_0|s)}[H_0]\mathbb{E}_{\pi_\theta(a_0|s)}[a_0]$, we obtain

$$\begin{aligned} L(s) &= \eta^\star \Sigma_\theta^{-1}(s)\phi_\theta(s) + \mathbb{E}_{\pi_\theta(a_0|s)}\left[\nabla_a \widehat{Q}(s, a)|_{a=a_0}\right] - \mathbb{E}_{\pi_\theta(a_0|s)}[H_0]\mathbb{E}_{\pi_\theta(a_0|s)}[a_0] \\ &= \eta^\star \Sigma_\theta^{-1}(s)\phi_\theta(s) + \mathbb{E}_{\pi_\theta(a_0|s)}\left[\nabla_a \widehat{Q}(s, a)|_{a=a_0}\right] - \mathbb{E}_{\pi_\theta(a_0|s)}[H_0]\phi_\theta(s) \\ &= (\eta^\star \Sigma_\theta^{-1}(s) - \mathbb{E}_{\pi_\theta(a_0|s)}[H_0(s)])\phi_\theta(s) + \mathbb{E}_{\pi_\theta(a_0|s)}\left[\nabla_a \widehat{Q}(s, a)|_{a=a_0}\right] \\ &= F(s)\phi_\theta(s) + \mathbb{E}_{\pi_\theta(a_0|s)}\left[\nabla_a \widehat{Q}(s, a)|_{a=a_0}\right]. \end{aligned} \tag{49}$$

Therefore, we have a second-order optimization step

$$\phi_+(s) = \phi_\theta(s) + F^{-1}(s)\mathbb{E}_{\pi_\theta(a_0|s)}\left[\nabla_a \widehat{Q}(s, a)|_{a=a_0}\right], \tag{50}$$

where $F^{-1}(s) = \eta^\star \Sigma_\theta^{-1}(s) - \mathbb{E}_{\pi_\theta(a_0|s)}[H_0(s)]$ is a curvature matrix. As described in the main paper, this interpretation is only valid when the equality $\mathbb{E}_{\pi_\theta(a_0|s)}[H_0(s)a_0(s)] = \mathbb{E}_{\pi_\theta(a_0|s)}[H_0]\mathbb{E}_{\pi_\theta(a_0|s)}[a_0]$ holds. While this equality does not hold in general, it holds when only one sample $a_0 \sim \pi_\theta(a_0|s)$ is used. Nonetheless, we can still use the expectation of Taylor's approximation to perform policy update regardless of this assumption.

A.3 PROOF OF GAUSS-NEWTON APPROXIMATION

Let $f(s, a) = \exp(\widehat{Q}(s, a))$, then the Hessian $H(s) = \nabla_a^2 \widehat{Q}(s, a)$ can be expressed as

$$\begin{aligned} H(s) &= \nabla_a\left[\nabla_a \log f(s, a)\right] \\ &= \nabla_a\left[\nabla_a f(s, a)f(s, a)^{-1}\right] \\ &= \nabla_a f(s, a)\left(\nabla_a f(s, a)^{-1}\right)^\top + \nabla_a^2 f(s, a)f(s, a)^{-1} \\ &= \nabla_a f(s, a)\nabla_f f(s, a)^{-1}\left(\nabla_a f(s, a)\right)^\top + \nabla_a^2 f(s, a)f(s, a)^{-1} \\ &= -\nabla_a f(s, a)f(s, a)^{-2}\left(\nabla_a f(s, a)\right)^\top + \nabla_a^2 f(s, a)f(s, a)^{-1} \\ &= -\left(\nabla_a f(s, a)f(s, a)^{-1}\right)\left(\nabla_a f(s, a)f(s, a)^{-1}\right)^\top + \nabla_a^2 f(s, a)f(s, a)^{-1} \\ &= -\nabla_a \log f(s, a)\nabla_a \log f(s, a)^\top + \nabla_a^2 f(s, a)f(s, a)^{-1} \\ &= -\nabla_a \widehat{Q}(s, a)\nabla_a \widehat{Q}(s, a)^\top + \nabla_a^2 \exp(\widehat{Q}(s, a))\exp(-\widehat{Q}(s, a)), \end{aligned} \tag{51}$$

which concludes the proof.

Beside Gauss-Newton approximation, an alternative approach is to impose a special structure on $\widehat{Q}(s, a)$ so that Hessians are always negative semi-definite. In literature, there exists two special structures that satisfies this requirement.

**Normalized advantage function** (NAF) (Gu et al., 2016): NAF represents the critic by a quadratic function with a negative curvature:

$$\widehat{Q}^{\mathrm{NAF}}(s, a) = \frac{1}{2}(a - b(s))^{\top} W(s)(a - b(s)) + V(s), \tag{52}$$

where a negative-definite matrix-valued function $W(s)$, a vector-valued function $b(s)$ and a scalar-valued function $V(s)$ are parameterized functions whose their parameters are learned by policy evaluation methods such as Q-learning (Sutton & Barto, 1998). With NAF, negative definite Hessians can be simply obtained as $H(s) = W(s)$. However, a significant disadvantage of NAF is that it assumes the action-value function is quadratic regardless of states and this is generally not true for most reward functions. Moreover, the Hessians become action-independent even though the critic is a function of actions.

**Input convex neural networks** (ICNNs) (Amos et al., 2017): ICNNs are neural networks with special structures which make them convex w.r.t. their inputs. Since Hessians of concave functions are always negative semi-definite, we may use ICNNs to represent a negative critic and directly use its Hessians. However, similarly to NAF, ICNNs implicitly assume that the action-value function is concave w.r.t. actions regardless of states and this is generally not true for most reward functions.

### A.4 Gradient of the Loss Functions for Learning Parameterized Actor

We first consider the weight mean-squared-error loss function where the guide actor is $\mathcal{N}(a|\phi_+(s), \Sigma_+(s))$ and the current actor is $\mathcal{N}(a|\phi_\theta(s), \Sigma_\theta(s))$. Taylor's approximation of $\widehat{Q}(s, a)$ at $a_0 = \phi_\theta(s)$ is

$$\widehat{Q}(s, a) = \frac{1}{2} a^{\top} H_{\phi_\theta}(s) a + a^{\top} \psi_{\phi_\theta}(s) + \xi_{\phi_\theta}(s). \tag{53}$$

By assuming that $H_{\phi_\theta}(s)$ is strictly negative definite[5], we can take a derivative of this approximation w.r.t. $a$ and set it to zero to obtain $a = H_{\phi_\theta}^{-1}(s)\nabla_a \widehat{Q}(s, a) - H_{\phi_\theta}^{-1}(s)\psi_{\phi_\theta}(s)$. Replacing $a$ by $\phi_\theta(s)$ and $\phi_+(s)$ yields

$$\phi_\theta(s) = H_{\phi_\theta}^{-1}(s)\nabla_a \widehat{Q}(s, a)|_{a=\phi_\theta(s)} - H_{\phi_\theta}^{-1}(s)\psi_{\phi_\theta}(s), \tag{54}$$

$$\phi_+(s) = H_{\phi_\theta}^{-1}(s)\nabla_a \widehat{Q}(s, a)|_{a=\phi_+(s)} - H_{\phi_\theta}^{-1}(s)\psi_{\phi_\theta}(s). \tag{55}$$

Recall that the weight mean-squared-error is defined as

$$\mathcal{L}^{\mathrm{W}}(\theta) = \mathbb{E}_{p^\beta(s)}\left[\|\phi_\theta(s) - \phi_+(s)\|_{F(s)}^2\right]. \tag{56}$$

---

[5]This can be done by subtracting a small positive value to the diagonal entries of Gauss-Newton approximation.

Then, we consider its gradient w.r.t. $\boldsymbol{\theta}$ as follows:

$$
\begin{aligned}
\nabla_{\boldsymbol{\theta}} \mathcal{L}^{\mathrm{W}}(\boldsymbol{\theta}) &= 2\mathbb{E}_{p^{\beta}(\boldsymbol{s})} \left[ \nabla_{\boldsymbol{\theta}} \boldsymbol{\phi}_{\boldsymbol{\theta}}(\boldsymbol{s}) \boldsymbol{F}(\boldsymbol{s}) \left( \boldsymbol{\phi}_{\boldsymbol{\theta}}(\boldsymbol{s}) - \boldsymbol{\phi}_{+}(\boldsymbol{s}) \right) \right] \\
&= 2\mathbb{E}_{p^{\beta}(\boldsymbol{s})} \left[ \nabla_{\boldsymbol{\theta}} \boldsymbol{\phi}_{\boldsymbol{\theta}}(\boldsymbol{s}) (\eta^{\star} \boldsymbol{\Sigma}_{\boldsymbol{\theta}}^{-1}(\boldsymbol{s}) - \boldsymbol{H}_{\boldsymbol{\phi}_{\boldsymbol{\theta}}}(\boldsymbol{s})) \left( \boldsymbol{\phi}_{\boldsymbol{\theta}}(\boldsymbol{s}) - \boldsymbol{\phi}_{+}(\boldsymbol{s}) \right) \right] \\
&= 2\mathbb{E}_{p^{\beta}(\boldsymbol{s})} \left[ \nabla_{\boldsymbol{\theta}} \boldsymbol{\phi}_{\boldsymbol{\theta}}(\boldsymbol{s}) (\eta^{\star} \boldsymbol{\Sigma}_{\boldsymbol{\theta}}^{-1}(\boldsymbol{s}) - \boldsymbol{H}_{\boldsymbol{\phi}_{\boldsymbol{\theta}}}(\boldsymbol{s})) \right. \\
&\qquad\qquad \left. \times \left( \boldsymbol{H}_{\boldsymbol{\phi}_{\boldsymbol{\theta}}}^{-1}(\boldsymbol{s}) \nabla_{\boldsymbol{a}} \widehat{Q}(\boldsymbol{s}, \boldsymbol{a})|_{\boldsymbol{a}=\boldsymbol{\phi}_{\boldsymbol{\theta}}(\boldsymbol{s})} - \boldsymbol{H}_{\boldsymbol{\phi}_{\boldsymbol{\theta}}}^{-1}(\boldsymbol{s}) \nabla_{\boldsymbol{a}} \widehat{Q}(\boldsymbol{s}, \boldsymbol{a})|_{\boldsymbol{a}=\boldsymbol{\phi}_{+}(\boldsymbol{s})} \right) \right] \\
&= 2\eta^{\star} \mathbb{E}_{p^{\beta}(\boldsymbol{s})} \left[ \nabla_{\boldsymbol{\theta}} \boldsymbol{\phi}_{\boldsymbol{\theta}}(\boldsymbol{s}) \boldsymbol{\Sigma}_{\boldsymbol{\theta}}^{-1}(\boldsymbol{s}) \boldsymbol{H}_{\boldsymbol{\phi}_{\boldsymbol{\theta}}}^{-1}(\boldsymbol{s}) \left( \nabla_{\boldsymbol{a}} \widehat{Q}(\boldsymbol{s}, \boldsymbol{a})|_{\boldsymbol{a}=\boldsymbol{\phi}_{\boldsymbol{\theta}}(\boldsymbol{s})} - \nabla_{\boldsymbol{a}} \widehat{Q}(\boldsymbol{s}, \boldsymbol{a})|_{\boldsymbol{a}=\boldsymbol{\phi}_{+}(\boldsymbol{s})} \right) \right] \\
&\quad + 2\mathbb{E}_{p^{\beta}(\boldsymbol{s})} \left[ \nabla_{\boldsymbol{\theta}} \boldsymbol{\phi}_{\boldsymbol{\theta}}(\boldsymbol{s}) \left( \nabla_{\boldsymbol{a}} \widehat{Q}(\boldsymbol{s}, \boldsymbol{a})|_{\boldsymbol{a}=\boldsymbol{\phi}_{+}(\boldsymbol{s})} - \nabla_{\boldsymbol{a}} \widehat{Q}(\boldsymbol{s}, \boldsymbol{a})|_{\boldsymbol{a}=\boldsymbol{\phi}_{\boldsymbol{\theta}}(\boldsymbol{s})} \right) \right] \\
&= -2\mathbb{E}_{p^{\beta}(\boldsymbol{s})} \left[ \nabla_{\boldsymbol{\theta}} \boldsymbol{\phi}_{\boldsymbol{\theta}}(\boldsymbol{s}) \nabla_{\boldsymbol{a}} \widehat{Q}(\boldsymbol{s}, \boldsymbol{a})|_{\boldsymbol{a}=\boldsymbol{\phi}_{\boldsymbol{\theta}}(\boldsymbol{s})} \right] + 2\mathbb{E}_{p^{\beta}(\boldsymbol{s})} \left[ \nabla_{\boldsymbol{\theta}} \boldsymbol{\phi}_{\boldsymbol{\theta}}(\boldsymbol{s}) \nabla_{\boldsymbol{a}} \widehat{Q}(\boldsymbol{s}, \boldsymbol{a})|_{\boldsymbol{a}=\boldsymbol{\phi}_{+}(\boldsymbol{s})} \right] \\
&\quad + 2\eta^{\star} \mathbb{E}_{p^{\beta}(\boldsymbol{s})} \left[ \nabla_{\boldsymbol{\theta}} \boldsymbol{\phi}_{\boldsymbol{\theta}}(\boldsymbol{s}) \boldsymbol{\Sigma}^{-1}(\boldsymbol{s}) \boldsymbol{H}_{\boldsymbol{\phi}_{\boldsymbol{\theta}}}^{-1}(\boldsymbol{s}) \nabla_{\boldsymbol{a}} \widehat{Q}(\boldsymbol{s}, \boldsymbol{a})|_{\boldsymbol{a}=\boldsymbol{\phi}_{\boldsymbol{\theta}}(\boldsymbol{s})} \right] \\
&\quad - 2\eta^{\star} \mathbb{E}_{p^{\beta}(\boldsymbol{s})} \left[ \nabla_{\boldsymbol{\theta}} \boldsymbol{\phi}_{\boldsymbol{\theta}}(\boldsymbol{s}) \boldsymbol{\Sigma}^{-1}(\boldsymbol{s}) \boldsymbol{H}_{\boldsymbol{\phi}_{\boldsymbol{\theta}}}^{-1}(\boldsymbol{s}) \nabla_{\boldsymbol{a}} \widehat{Q}(\boldsymbol{s}, \boldsymbol{a})|_{\boldsymbol{a}=\boldsymbol{\phi}_{+}(\boldsymbol{s})} \right].
\end{aligned}
\tag{57}
$$

This concludes the proof for the gradient in Eq.(28). Note that we should not directly replace the mean functions in the weight mean-square-error by Eq.(54) and Eq.(55) before expanding the gradient. This is because the analysis would require computing the gradient w.r.t. $\boldsymbol{\theta}$ inside the Hessians and this is not trivial. Moreover, when we perform gradient descent in practice, the mean $\boldsymbol{\phi}_{+}(\boldsymbol{s})$ is considered as a constant w.r.t. $\boldsymbol{\theta}$ similarly to an output function in supervised learning.

The gradient for the mean-squared-error can be derived similarly. Let the mean-squared-error be defined as

$$
\mathcal{L}^{\mathrm{M}}(\boldsymbol{\theta}) = \frac{1}{2} \mathbb{E}_{p^{\beta}(\boldsymbol{s})} \left[ \| \boldsymbol{\phi}_{\boldsymbol{\theta}}(\boldsymbol{s}) - \boldsymbol{\phi}_{+}(\boldsymbol{s}) \|_2^2 \right].
\tag{58}
$$

Its gradient w.r.t. $\boldsymbol{\theta}$ is given by

$$
\begin{aligned}
\nabla_{\boldsymbol{\theta}} \mathcal{L}^{\mathrm{M}}(\boldsymbol{\theta}) &= \mathbb{E}_{p^{\beta}(\boldsymbol{s})} \left[ \nabla_{\boldsymbol{\theta}} \boldsymbol{\phi}_{\boldsymbol{\theta}}(\boldsymbol{s}) \left( \boldsymbol{\phi}_{\boldsymbol{\theta}}(\boldsymbol{s}) - \boldsymbol{\phi}_{+}(\boldsymbol{s}) \right) \right] \\
&= \mathbb{E}_{p^{\beta}(\boldsymbol{s})} \left[ \nabla_{\boldsymbol{\theta}} \boldsymbol{\phi}_{\boldsymbol{\theta}}(\boldsymbol{s}) \boldsymbol{H}^{-1}(\boldsymbol{s}) \left( \nabla_{\boldsymbol{a}} \widehat{Q}(\boldsymbol{s}, \boldsymbol{a})|_{\boldsymbol{a}=\boldsymbol{\phi}_{\boldsymbol{\theta}}(\boldsymbol{s})} - \nabla_{\boldsymbol{a}} \widehat{Q}(\boldsymbol{s}, \boldsymbol{a})|_{\boldsymbol{a}=\boldsymbol{\phi}_{+}(\boldsymbol{s})} \right) \right],
\end{aligned}
\tag{59}
$$

which concludes the proof for the gradient in Eq.(30).

To show that $\nabla_{\boldsymbol{a}} \widehat{Q}(\boldsymbol{s}, \boldsymbol{a})|_{\boldsymbol{a}=\boldsymbol{\phi}_{+}(\boldsymbol{s})} = 0$ when $\eta^{\star} = 0$, we directly substitute $\eta^{\star} = 0$ into $\boldsymbol{\phi}_{+}(\boldsymbol{s}) = \left( \eta \boldsymbol{\Sigma}^{-1}(\boldsymbol{s}) - \boldsymbol{H}_{\boldsymbol{\phi}_{\boldsymbol{\theta}}}(\boldsymbol{s}) \right)^{-1} \left( \eta^{\star} \boldsymbol{\Sigma}^{-1}(\boldsymbol{s}) \boldsymbol{\phi}_{\boldsymbol{\theta}}(\boldsymbol{s}) + \boldsymbol{\psi}_{\boldsymbol{\phi}_{\boldsymbol{\theta}}}(\boldsymbol{s}) \right)$ and this yield

$$
\boldsymbol{\phi}_{+}(\boldsymbol{s}) = -\boldsymbol{H}_{\boldsymbol{\phi}_{\boldsymbol{\theta}}}^{-1}(\boldsymbol{s}) \boldsymbol{\psi}_{\boldsymbol{\phi}_{\boldsymbol{\theta}}}(\boldsymbol{s}).
\tag{60}
$$

Since $\boldsymbol{\phi}_{+}(\boldsymbol{s}) = \boldsymbol{H}_{\boldsymbol{\phi}_{\boldsymbol{\theta}}}^{-1}(\boldsymbol{s}) \nabla_{\boldsymbol{a}} \widehat{Q}(\boldsymbol{s}, \boldsymbol{a})|_{\boldsymbol{a}=\boldsymbol{\phi}_{+}(\boldsymbol{s})} - \boldsymbol{H}_{\boldsymbol{\phi}_{\boldsymbol{\theta}}}^{-1}(\boldsymbol{s}) \boldsymbol{\psi}_{\boldsymbol{\phi}_{\boldsymbol{\theta}}}(\boldsymbol{s})$ from Eq.(55) and the Hessians are non-zero, it has to be that $\nabla_{\boldsymbol{a}} \widehat{Q}(\boldsymbol{s}, \boldsymbol{a})|_{\boldsymbol{a}=\boldsymbol{\phi}_{+}(\boldsymbol{s})} = 0$. This is intuitive since without the KL constraint, the mean of the guide actor always be at the optima of the second-order Taylor's approximation and the gradients are zero at the optima.

## A.5 Relation to Q-learning with Normalized Advantage Function

The *normalized advantage function* (NAF) (Gu et al., 2016) is defined as

$$
\widehat{Q}^{\mathrm{NAF}}(\boldsymbol{s}, \boldsymbol{a}) = \frac{1}{2} (\boldsymbol{a} - \boldsymbol{b}(\boldsymbol{s}))^{\top} \boldsymbol{W}(\boldsymbol{s}) (\boldsymbol{a} - \boldsymbol{b}(\boldsymbol{s})) + V(\boldsymbol{s}),
\tag{61}
$$

where $\boldsymbol{W}(\boldsymbol{s})$ is a negative definite matrix. Gu et al. (2016) proposed to perform Q-learning with NAF by using the fact that $\operatorname{argmax}_{\boldsymbol{a}} \widehat{Q}^{\mathrm{NAF}}(\boldsymbol{s}, \boldsymbol{a}) = \boldsymbol{b}(\boldsymbol{s})$ and $\max_{\boldsymbol{a}} \widehat{Q}^{\mathrm{NAF}}(\boldsymbol{s}, \boldsymbol{a}) = V(\boldsymbol{s})$.

Here, we show that GAC includes the Q-learning method by Gu et al. (2016) as its special case. This can be shown by using NAF as a critic instead of performing Taylor's approximation of the critic.

Firstly, we expand the quadratic term of NAF as follows:

$$\widehat{Q}^{\mathrm{NAF}}(\boldsymbol{s},\boldsymbol{a}) = \frac{1}{2}(\boldsymbol{a}-\boldsymbol{b}(\boldsymbol{s}))^\top \boldsymbol{W}(\boldsymbol{s})(\boldsymbol{a}-\boldsymbol{b}(\boldsymbol{s})) + V(\boldsymbol{s})$$

$$= \frac{1}{2}\boldsymbol{a}^\top \boldsymbol{W}(\boldsymbol{s})\boldsymbol{a} - \boldsymbol{a}^\top \boldsymbol{W}(\boldsymbol{s})\boldsymbol{b}(\boldsymbol{s}) + \frac{1}{2}\boldsymbol{b}(\boldsymbol{s})^\top \boldsymbol{W}(\boldsymbol{s})\boldsymbol{b}(\boldsymbol{s}) + V(\boldsymbol{s})$$

$$= \frac{1}{2}\boldsymbol{a}^\top \boldsymbol{W}(\boldsymbol{s})\boldsymbol{a} + \boldsymbol{a}^\top \boldsymbol{\psi}(\boldsymbol{s}) + \xi(\boldsymbol{s}), \tag{62}$$

where $\boldsymbol{\psi}(\boldsymbol{s}) = -\boldsymbol{W}(\boldsymbol{s})\boldsymbol{b}(\boldsymbol{s})$ and $\xi(\boldsymbol{s}) = \frac{1}{2}\boldsymbol{b}(\boldsymbol{s})^\top \boldsymbol{W}(\boldsymbol{s})\boldsymbol{b}(\boldsymbol{s}) + V(\boldsymbol{s})$. By substituting the quadratic model obtained by NAF into the GAC framework, the guide actor is now given by $\tilde{\pi}(\boldsymbol{a}|\boldsymbol{s}) = \mathcal{N}(\boldsymbol{a}|\boldsymbol{\phi}_+(\boldsymbol{s}), \boldsymbol{\Sigma}_+(\boldsymbol{s})))$ with

$$\boldsymbol{\phi}_+(\boldsymbol{s}) = (\eta^\star \boldsymbol{\Sigma}^{-1}(\boldsymbol{s})) - \boldsymbol{W}(\boldsymbol{s}))^{-1}(\eta^\star \boldsymbol{\Sigma}^{-1}(\boldsymbol{s})\boldsymbol{\phi}_{\boldsymbol{\theta}}(\boldsymbol{s}) - \boldsymbol{W}(\boldsymbol{s})\boldsymbol{b}(\boldsymbol{s})) \tag{63}$$

$$\boldsymbol{\Sigma}_+(\boldsymbol{s}) = (\eta^\star + \omega^\star)(\eta^\star \boldsymbol{\Sigma}^{-1}(\boldsymbol{s})) - \boldsymbol{W}(\boldsymbol{s})). \tag{64}$$

To obtain Q-learning with NAF, we set $\eta^\star = 0$, i.e., we perform a greedy maximization where the KL upper-bound approaches infinity, and this yields

$$\boldsymbol{\phi}_+(\boldsymbol{s}) = -\boldsymbol{W}(\boldsymbol{s})^{-1}(-\boldsymbol{W}(\boldsymbol{s})\boldsymbol{b}(\boldsymbol{s}))$$

$$= \boldsymbol{b}(\boldsymbol{s}), \tag{65}$$

which is the policy obtained by performing Q-learning with NAF. Thus, NAF with Q-learning is a special case of GAC if Q-learning is also used in GAC to learn the critic.

## B  PSEUDO-CODE OF GAC

The pseudo-code of GAC is given in Algorithm 1. The source code is available at `https://github.com/voot-t/guide-actor-critic`.

## C  EXPERIMENT DETAILS

### C.1  IMPLEMENTATION

We try to follow the network architecture proposed by the authors of each baseline method as close as possible. For GAC and DDPG, we use neural networks with two hidden layers for the actor network and the critic network. For both networks the first layer has 400 hidden units and the second layer has 300 units. For NAF, we use neural networks with two hidden layers to represent each of the functions $\boldsymbol{b}(\boldsymbol{s})$, $\boldsymbol{W}(\boldsymbol{s})$ and $V(\boldsymbol{s})$ where each layer has 200 hidden units. All hidden units use the `relu` activation function except for the output of the actor network where we use the `tanh` activation function to bound actions. We use the Adam optimizer (Kingma & Ba, 2014) with learning rate 0.001 and 0.0001 for the critic network and the actor network, respectively. The moving average step for target networks is set to $\tau = 0.001$. The maximum size of the replay buffer is set to 1000000. The mini-batches size is set to $N = 256$. The weights of the actor and critic networks are initialized as described by Glorot & Bengio (2010), except for the output layers where the initial weights are drawn uniformly from $\mathrm{U}(-0.003, 0.003)$, as described by Lillicrap et al. (2015). The initial covariance $\boldsymbol{\Sigma}$ in GAC is set to be an identity matrix. DDPG and QNAF use the OU-process with noise parameters $\theta = 0.15$ and $\sigma = 0.2$ for exploration .

For TRPO, we use the implementation publicly available at `https://github.com/openai/baselines`. We also use the provided network architecture and hyper-parameters except the batch size where we use 1000 instead of 1024 since this is more suitable in our test setup.

For GAC, the KL upper-bound is fixed to $\epsilon = 0.0001$. The entropy lower-bound $\kappa$ is adjusted heuristically by

$$\kappa = \max(0.99(E - E_0) + E_0, E_0), \tag{71}$$

where $E \approx \mathbb{E}_{p^\beta(\boldsymbol{s})}\left[\mathrm{H}(\pi_{\boldsymbol{\theta}}(\boldsymbol{a}|\boldsymbol{s}))\right]$ denotes the expected entropy of the current policy and $E_0$ denotes the entropy of a base policy $\mathcal{N}(\boldsymbol{a}|\boldsymbol{0}, 0.01\boldsymbol{I})$. This heuristic ensures that the lower-bound gradually decreases but the lower-bound cannot be too small. We apply this heuristic update once every 5000 training steps. The dual function is minimize by the sequential least-squares quadratic programming (SLSQP) method with an initial values $\eta = 0.05$ and $\omega = 0.05$. The number of samples for computing the target critic value is $M = 10$.

---

**Algorithm 1** Guide actor critic

---

1: **Input:** Initial actor $\pi_{\boldsymbol{\theta}}(\boldsymbol{a}|\boldsymbol{s}) = \mathcal{N}(\boldsymbol{a}|\boldsymbol{\phi}_{\boldsymbol{\theta}}(\boldsymbol{s}), \boldsymbol{\Sigma})$, critic $\widehat{Q}_{\boldsymbol{\nu}}(\boldsymbol{s}, \boldsymbol{a})$, target critic network $\widehat{Q}_{\bar{\boldsymbol{\nu}}}(\boldsymbol{s}, \boldsymbol{a})$, KL bound $\epsilon$, entropy bound $\kappa$, learning rates $\alpha_1, \alpha_2$, and data buffer $\mathcal{D} = \varnothing$.
2: **for** $t = 1, \dots, T_{\max}$ **do**
3:     **procedure** COLLECT TRANSITION SAMPLE
4:         Observe state $\boldsymbol{s}_t$ and sample action $\boldsymbol{a}_t \sim \mathcal{N}(\boldsymbol{a}|\boldsymbol{\phi}_{\boldsymbol{\theta}}(\boldsymbol{s}_t), \boldsymbol{\Sigma})$.
5:         Execute $\boldsymbol{a}_t$, receive reward $r_t$ and next state $\boldsymbol{s}'_t$.
6:         Add transition $\{(\boldsymbol{s}_t, \boldsymbol{a}_t, r_t, \boldsymbol{s}'_t)\}$ to $\mathcal{D}$.
7:     **end procedure**
8:     **procedure** LEARN
9:         Sample $N$ mini-batch samples $\{(\boldsymbol{s}_n, \boldsymbol{a}_n, r_n, \boldsymbol{s}'_n)\}_{n=1}^N$ uniformly from $\mathcal{D}$.
10:         **procedure** UPDATE CRITIC
11:             Sample actions $\{\boldsymbol{a}'_{n,m}\}_{m=1}^M \sim \mathcal{N}(\boldsymbol{a}|\boldsymbol{\phi}_{\boldsymbol{\theta}}(\boldsymbol{s}_n), \boldsymbol{\Sigma})$ for each $\boldsymbol{s}_n$.
12:             Compute $y_n$, update $\boldsymbol{\nu}$ by, e.g., Adam, and update $\bar{\boldsymbol{\nu}}$ by moving average:

$$y_n = r_n + \gamma \frac{1}{M} \sum_{m=1}^M \widehat{Q}_{\bar{\boldsymbol{\nu}}}(\boldsymbol{s}'_n, \boldsymbol{a}'_{n,m}), \tag{66}$$

$$\boldsymbol{\nu} \leftarrow \boldsymbol{\nu} - \alpha_1 \frac{1}{N} \sum_{n=1}^N \nabla_{\boldsymbol{\nu}} \left( \widehat{Q}_{\boldsymbol{\nu}}(\boldsymbol{s}_n, \boldsymbol{a}_n) - y_n \right)^2, \tag{67}$$

$$\bar{\boldsymbol{\nu}} \leftarrow \tau \boldsymbol{\nu} + (1 - \tau) \bar{\boldsymbol{\nu}}. \tag{68}$$

13:         **end procedure**
14:         **procedure** LEARN GUIDE ACTOR
15:             Compute $\boldsymbol{a}_{n,0}$ for each $\boldsymbol{s}_n$ by $\boldsymbol{a}_{n,0} = \boldsymbol{\phi}_{\boldsymbol{\theta}}(\boldsymbol{s}_n)$ or $\boldsymbol{a}_{n,0} \sim \mathcal{N}(\boldsymbol{a}|\boldsymbol{\phi}_{\boldsymbol{\theta}}(\boldsymbol{s}_n), \boldsymbol{\Sigma})$.
16:             Compute $\boldsymbol{g}_0(\boldsymbol{s}) = \nabla_{\boldsymbol{a}}\widehat{Q}(\boldsymbol{s}_n, \boldsymbol{a})|_{\boldsymbol{a}=\boldsymbol{a}_{n,0}}$ and $\boldsymbol{H}_0(\boldsymbol{s}_s) = -\boldsymbol{g}_0(\boldsymbol{s}_n)\boldsymbol{g}_0(\boldsymbol{s}_n)^\top$.
17:             Solve for $(\eta^\star, \omega^\star) = \operatorname{argmin}_{\eta>0, \omega>0} \widehat{g}(\eta, \omega)$ by a non-linear optimization method.
18:             Compute the guide actor $\widetilde{\pi}(\boldsymbol{a}|\boldsymbol{s}_n) = \mathcal{N}(\boldsymbol{a}|\boldsymbol{\phi}_+(\boldsymbol{s}_n), \boldsymbol{\Sigma}_+(\boldsymbol{s}_n))$ for each $\boldsymbol{s}_n$.
19:         **end procedure**
20:         **procedure** UPDATE PARAMETERIZED ACTOR
21:             Update policy parameter by, e.g., Adam, to minimize the MSE:

$$\boldsymbol{\theta} \leftarrow \boldsymbol{\theta} - \alpha_2 \frac{1}{N} \sum_{n=1}^N \nabla_{\boldsymbol{\theta}} \|\boldsymbol{\phi}_{\boldsymbol{\theta}}(\boldsymbol{s}_n) - \boldsymbol{\phi}_+(\boldsymbol{s}_n)\|_2^2. \tag{69}$$

22:             Update policy covariance by averaging the guide covariances:

$$\boldsymbol{\Sigma} \leftarrow \frac{1}{N} \boldsymbol{\Sigma}_+(\boldsymbol{s}_n). \tag{70}$$

23:         **end procedure**
24:     **end procedure**
25: **end for**
26: **Output:** Learned actor $\pi_{\boldsymbol{\theta}}(\boldsymbol{a}|\boldsymbol{s})$.

---

## C.2 ENVIRONMENTS AND RESULTS

We perform experiments on the OpenAI gym platform (Brockman et al., 2016) with Mujoco Physics simulator (Todorov et al., 2012) where all environments are v1. We use the state space, action space and the reward function as provided and did not perform any normalization or gradient clipping. The maximum time horizon in each episode is set to 1000. The discount factor $\gamma = 0.99$ is only used for learning and the test returns are computed without it.

Experiments are repeated for 10 times with different random seeds. The total computation time are reported in Table 1. The figures below show the results averaged over 10 trials. The y-axis indicates the averaged test returns where the test returns in each trial are computed once every 5000 training time steps by executing 10 test episodes without exploration. The error bar indicates standard error.

Table 1: The total computation time for training the policy for 1 million steps (0.1 million steps for the Invert-Pendulum task). The mean and standard error are computed over 10 trials with the unit in hours. TRPO is not included since it performs a lesser amount of update using batch data samples.

| Task | GAC-1 | GAC-0 | DDPG | QNAF |
|---|---|---|---|---|
| Inv-Pend. | 1.13(0.09) | 0.80(0.04) | 0.45(0.02) | 0.40(0.03) |
| Inv-Double-Pend. | 15.56(0.93) | 15.67(0.77) | 9.04(0.29) | 7.47(0.22) |
| Reacher | 17.23(0.87) | 12.64(0.38) | 10.67(0.37) | 30.91(2.09) |
| Swimmer | 12.43(0.74) | 11.94(0.74) | 9.61(0.52) | 32.44(2.21) |
| Half-Cheetah | 29.82(1.76) | 32.64(1.41) | 10.13(0.41) | 27.94(2.37) |
| Ant | 37.84(1.80) | 40.57(3.06) | 9.75(0.37) | 27.09(0.94) |
| Hopper | 18.99(1.06) | 14.22(0.56) | 8.74(0.45) | 26.48(1.42) |
| Walker2D | 33.42(0.96) | 31.71(1.84) | 7.92(0.11) | 26.94(1.99) |
| Humanoid | 111.07(2.99) | 106.80(6.80) | 13.90(1.60) | 30.43(2.21) |

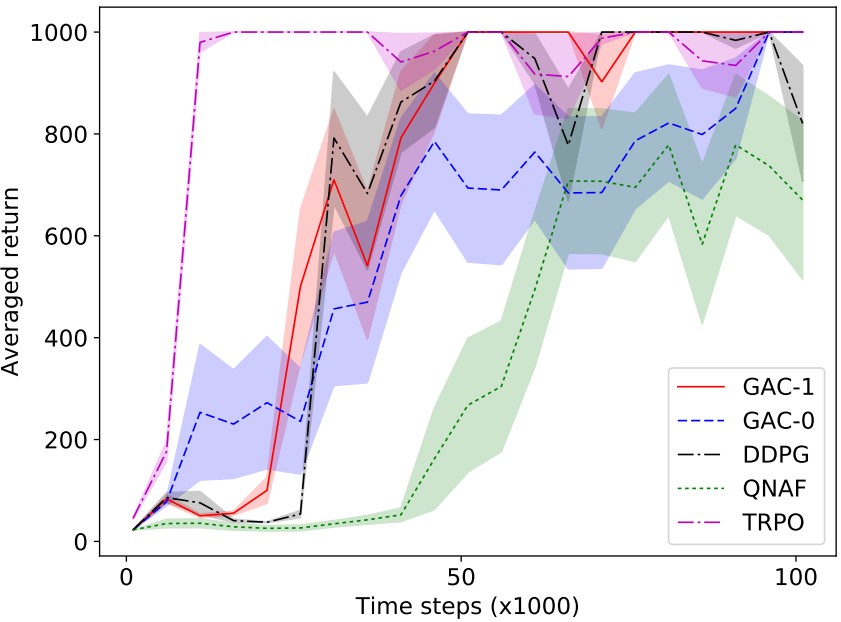

Figure 2: Performance averaged over 10 trials on the Inverted Pendulum task.

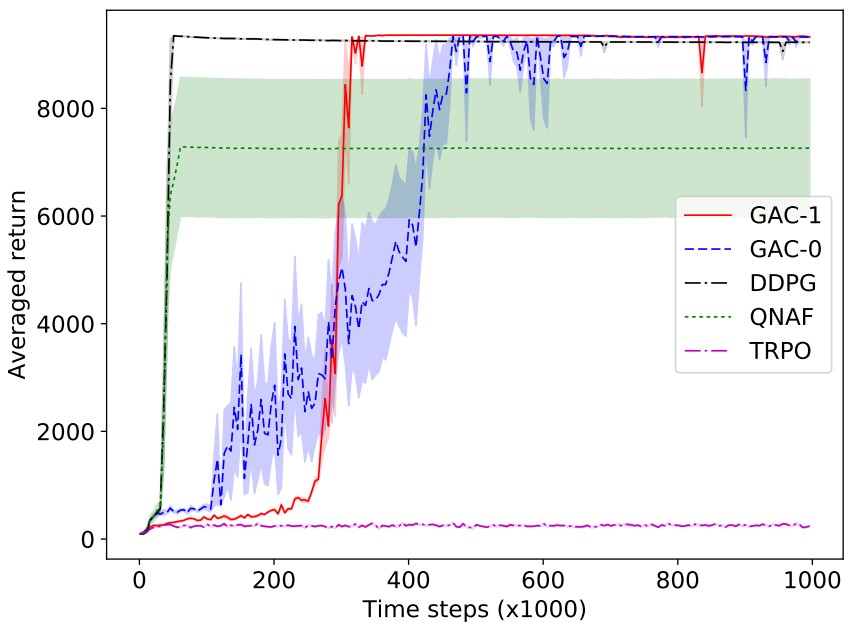

Figure 3: Performance averaged over 10 trials on the Inverted Double Pendulum task.

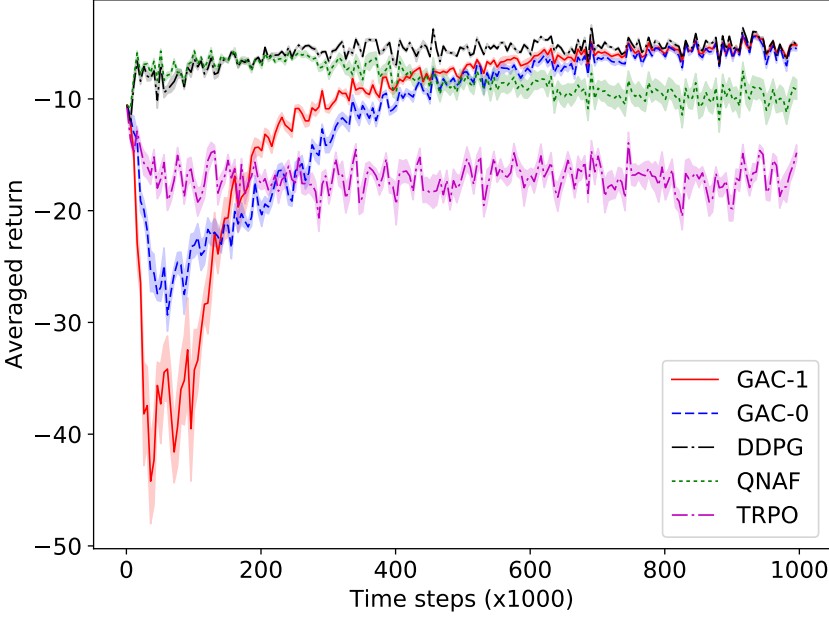

Figure 4: Performance averaged over 10 trials on the Reacher task.

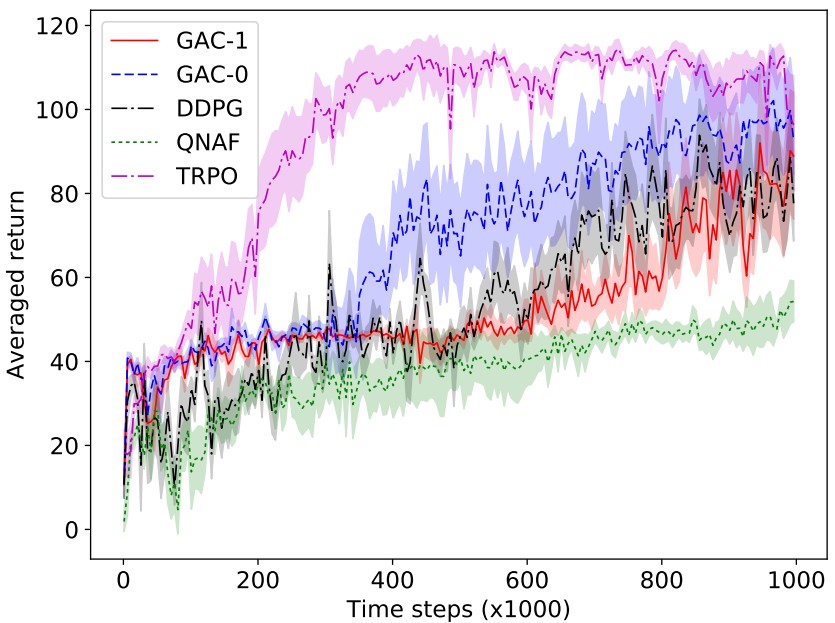

Figure 5: Performance averaged over 10 trials on the Swimmer task.

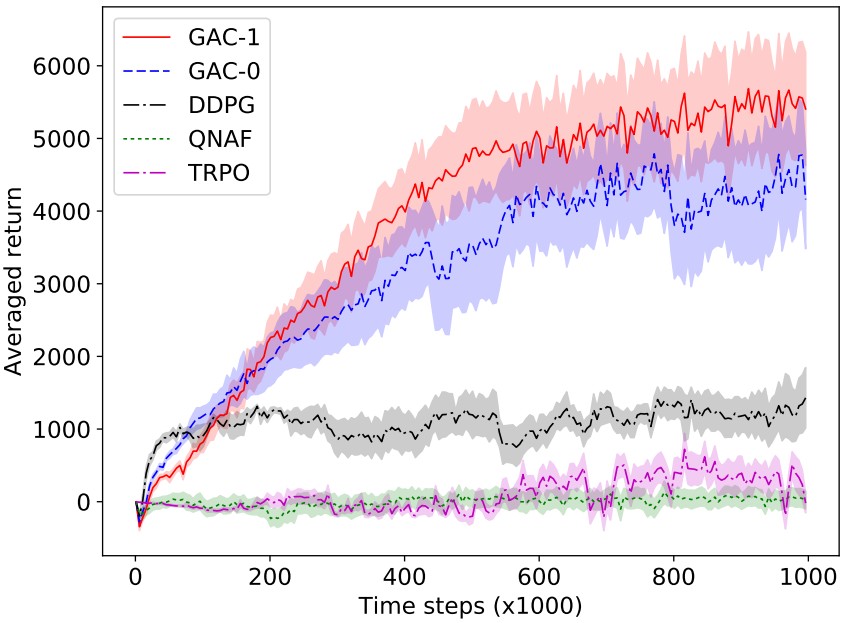

Figure 6: Performance averaged over 10 trials on the Half-Cheetah task.

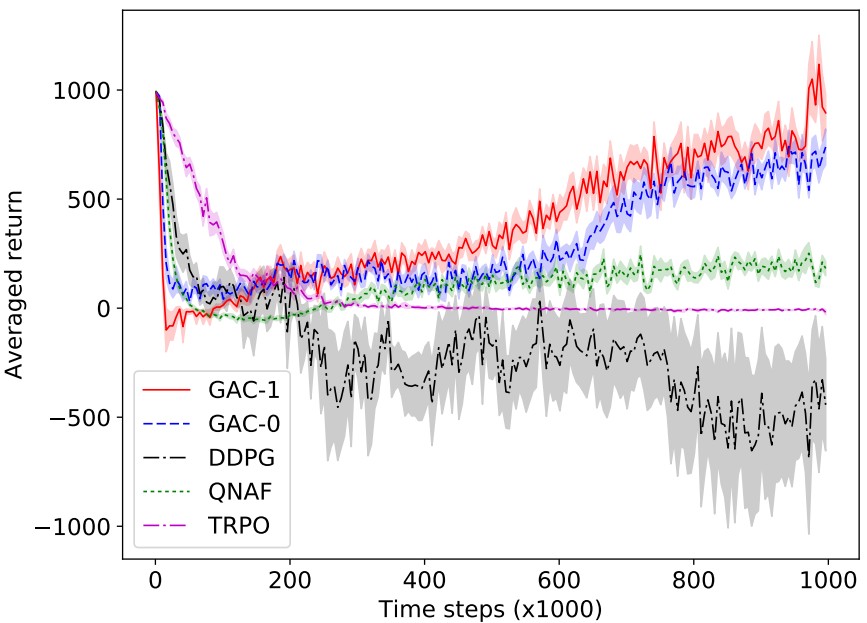

Figure 7: Performance averaged over 10 trials on the Ant task.

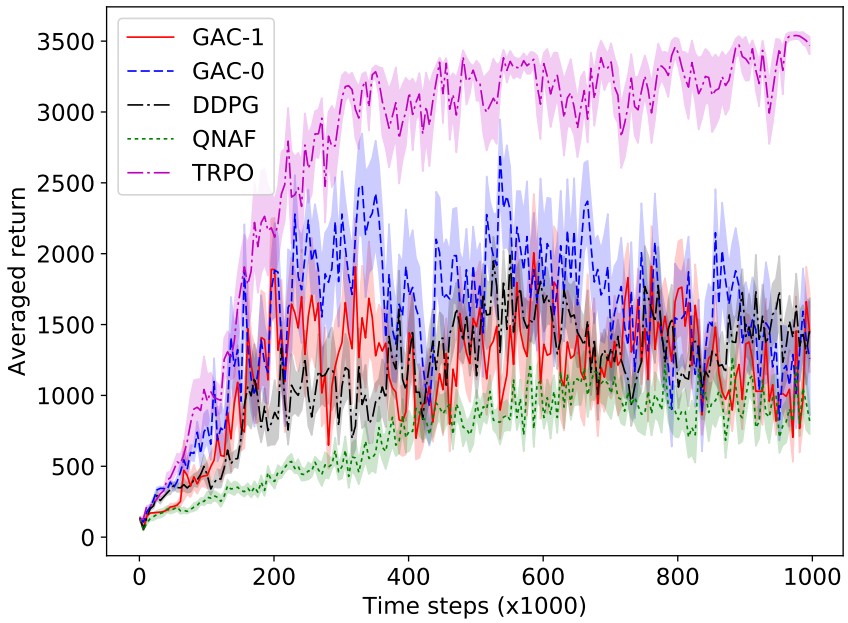

Figure 8: Performance averaged over 10 trials on the Hopper task.

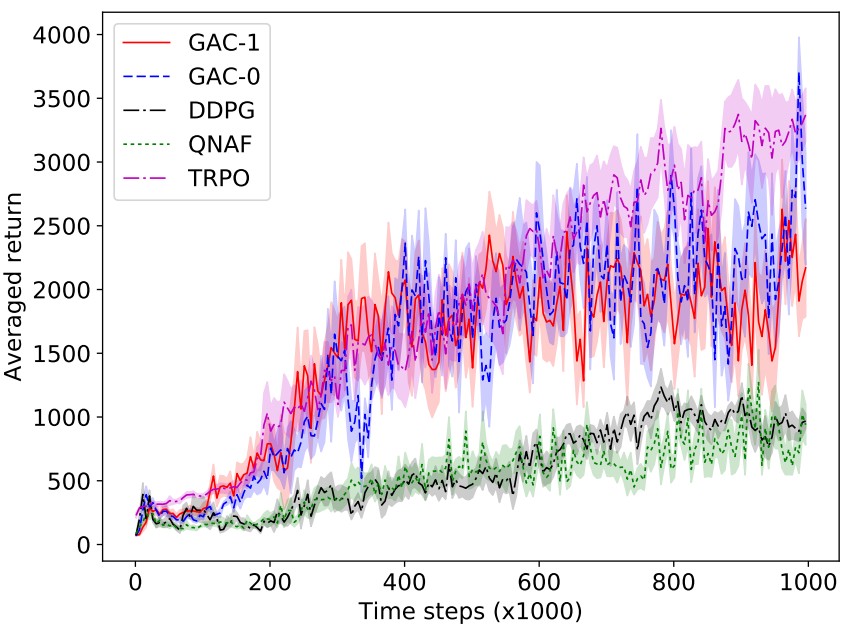

Figure 9: Performance averaged over 10 trials on the Walker2D task.

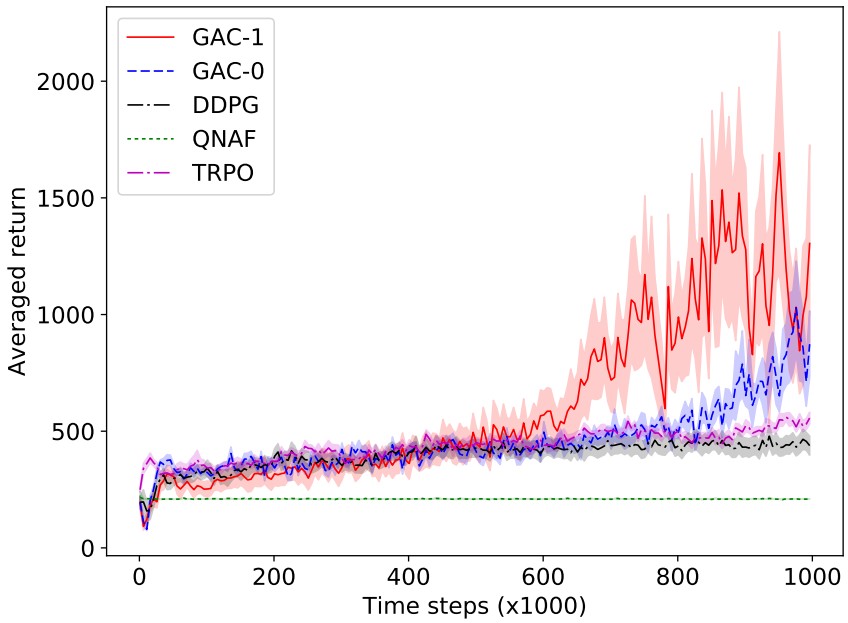

Figure 10: Performance averaged over 10 trials on the Humanoid task.

