# OpenReview forum: "Guide Actor-Critic for Continuous Control"
_ICLR.cc/2018/Conference — Accept (Poster)_

### Official Review · AnonReviewer2 · 2017-11-27
**A promising technique, using hessian of the critic, for learning the actor, for continuous control, with competitive results**

**Rating:** 6
**Confidence:** 2

**Review:**



The authors devise and explore use of the hessian of the
(approximate/learned) value function (the critic) to update the policy
(actor) in the actor-critic  approach to RL.  They connect their
technique, 'guide actor-critic' or GAC, to existing actor-critic
methods (authors claim only two published work use 1st order
information on the critic). They show that the 2nd order information
can be useful (in several of the 9 tasks, their GAC techniques were
best or competitive, and in only one, performed poorly compared to best).

The paper has a technical focus.

pros:

- Strict generalization of an existing (up to 1st order) actor-critic approaches.

- Compared to many existing techniques, on 9 tasks

cons:

- no mention of time costs, except that for more samples, S > 1, for
 taylor approximation, it can be very expensive.

- one would expect more information to strictly improve performance,
  but the results are a bit mixed (perhaps due to convergence to local
  optima and both actor and critic being learned at same time,
  or the Gaussian assumptions, etc.).

- relevance: the work presents a new approach to actor-critique strategy for
  reinforcement learning, remotely related to 'representation
  learning' (unless value and policies are deemed a form of
  representation).


Other comments/questions:

- Why does the performance start high on Ant (1000), then goes to 0
(all approaches)?

- How were the tasks selected? Are they all the continuous control
  tasks available in open ai?

---

> ### Author Response · Authors · 2017-12-11
> **Response to review 2**
>
> Thank you for your constructive review.  We address the reviewer's questions and comments below.
>
> 2.1: No mention of time costs, except that for more samples, S > 1, for taylor approximation, it can be very expensive.
> - We will include a table reporting the training time. Please see also our response to 1.4.
>
> 2.2: One would expect more information to strictly improve performance, but the results are a bit mixed.
> - The second-order information may not provide much benefit in simple tasks such as Inverted-pendulum, Inverted-double-pendulum and Reacher. However, in locomotion tasks except Hopper, there are significant differences in performances between the second-order method (GAC) and the first-order method (DPG). To make the results more convincing, we will increase the number of experiment trials to 10.
>
> 2.3: Why does the performance start high on Ant (1000), then goes to 0 (all approaches)?
> - The reward function in the Ant task is r(s,a) = forward_movement + survive_reward - control_cost - contact_cost. The survive reward is 1 if the agent does not fall over and 0 otherwise. All methods are initialized so that initial actions are close to 0, and this makes the agent moves only slightly and survives for the entire episodes (1000 steps). With this initial configuration, the performance sharply drops since random actions from exploration can yield high control costs while making the agent falls over. This behavior does not appear in other locomotion tasks since actions close to 0 will make the agent falls over and obtains low rewards at the beginning.
>
> 2.4: How were the tasks selected? Are they all the continuous control tasks available in open ai?
> - Classical continuous control tasks such as Pendulum and Mountain-Car are also available from OpenAI gym. The tasks we selected are considered more challenging than these classical tasks and are commonly used as benchmark tasks in recent literature.

---

### Official Review · AnonReviewer1 · 2017-11-27

**Rating:** 4
**Confidence:** 4

**Review:**

The paper introduces a modified actor-critic algorithm where a “guide actor” uses approximate second order methods to aid computation. The experimental results are similar to previously proposed methods.

The paper is fairly well-written, provides proofs of detailed properties of the algorithm, and has decent experimental results. However, the method is not properly motivated. As far as I can tell, the paper never answers the questions: Why do we need a guide actor? What problem does the guide actor solve?

The paper argues that the guide actor allows to introduce second order methods, but (1) there are other ways of doing so and (2) it’s not clear why we should want to use second-order methods in reinforcement learning in the first place. Using second order methods is not an end in itself. The experimental results show the authors have found a way to use second order methods without making performance *worse*. Given the high variability of deep RL, they have not convincingly shown it performs better.

The paper does not discuss the computational cost of the method. How does it compare to other methods? My worry is that the method is more complicated and slower than existing methods, without significantly improved performance.

I recommend the authors take the time to make a much stronger conceptual and empirical case for their algorithm.

---

> ### Author Response · Authors · 2017-12-11
> **Response to review 1**
>
> Thank you for your constructive review.  We address the reviewer's questions and comments below.
>
> 1.1: The method is not properly motivated. It’s not clear why we should want to use second-order methods in reinforcement learning in the first place.
> - Second-order methods leverage curvature information for optimization and this often leads to faster learning when compared to first-order methods. In the actor-critic framework, a similar idea was pursued by natural actor-critic and TRPO where curvature information is given by the Fisher information matrix. More recently, an approximate Newton’s method in the policy search framework was also proposed by Furmston et al. (2016). These methods have established that second-order methods can significantly outperforms first-order methods in RL.
>
> 1.2: There are other ways of doing second-order methods. Why do we need a guide actor? What problem does the guide actor solve?
> - It is true that second-order methods can be applied without the guide actor. However, they are infeasible in deep RL since the size of the Hessian matrix or the Fisher information matrix of the RL objective in Eq.(3) depends on the number of the policy parameter. Existing methods avoid this issue by a diagonal approximation or a factor approximation. Clearly, these approximations lead to a loss of useful curvature information. By reformulating the problem and optimizing the guide actor in the action space, we obtain a closed-form second-order update that utilizes a full Hessian matrix of the critic. Moreover, our second-order update incorporates both the KL and entropy constraints, and we believe this is new in deep RL setting.
>
> 1.3: Given the high variability of deep RL, they have not convincingly shown it performs better.
> - We will increase the number of experiment trials to 10 to make the results more convincing.
>
> 1.4: How does the computation time compare to other methods? My worry is that the method is more complicated and slower than existing methods, without significantly improved performance.
> - We will include a table reporting the training time of each method. We agree that our method is computationally more expensive than other methods due to the inner optimization for finding eta and omega, but this cost can be reduced by letting eta and omega be external tuning parameters. Without the inner optimization, the computation cost of our method for S <= 1 is comparable to that of the first order method since we use Gauss-Newton approximation where the outer product operation is computationally cheap.

---

### Official Review · AnonReviewer3 · 2017-12-03
**Clever approach**

**Rating:** 7
**Confidence:** 4

**Review:**

The paper presents a clever trick for updating the actor in an actor-critic setting: computing a guide actor that diverges from the actor to improve critic value, then updating the actor parameters towards the guide actor. This can be done since, when the parametrized actor is Gaussian and the critic value can be well-approximated as quadratic in the action, the guide actor can be optimized in closed form.

The paper is mostly clear and well-presented, except for two issues: 1) there is virtually nothing novel presented in the first half of the paper (before Section 3.3); and 2) the actual learning step is only presented on page 6, making it hard to understand the motivation behind the guide actor until very late through the paper.

The presented method itself seems to be an important contribution, even if the results are not overwhelmingly positive. It'd be interesting to see a more elaborate analysis of why it works well in some domains but not in others. More trials are also needed to alleviate any suspicion of lucky trials.

There are some other issues with the presentation of the method, but these don't affect the merit of the method:

1. Returns are defined from an initial distribution that is stationary for the policy. While this makes sense in well-mixing domains, the experiment domains are not well-mixing for most policies during training, for example a fallen humanoid will not get up on its own, and must be reset.

2. The definition of beta(a|s) as a mixture of past actors is inconsistent with the sampling method, which seems to be a mixture of past trajectories.

3. In the first paragraph of Section 3.3: "[...] the quality of a guide actor mostly depends on the accuracy of Taylor's approximation." What else does it depend on? Then: "[...] the action a_0 should be in a local vicinity of a."; and "[...] the action a_0 should be similar to actions sampled from pi_theta(a|s)." What do you mean "should"? In order for the Taylor approximation to be good?

4. The line before (19) is confusing, since (19) is exact and not an approximation. For the approximation (20), it isn't clear if this is a good approximation. Why/when is the 2nd term in (19) small?

5. The parametrization nu of \hat{Q} is never specified in Section 3.6. This is important in order to evaluate the complexities involved in computing its Hessian.

---

> ### Author Response · Authors · 2017-12-11
> **Response to review 3**
>
> Thank you for your constructive review.  We address the reviewer's questions and comments below.
>
> 3.1: There is virtually nothing novel presented in the first half of the paper (before Section 3.3). 2) the actual learning step is only presented on page 6, making it hard to understand the motivation behind the guide actor until very late through the paper.
> - We expect the new subsection (please see our "Author response" comment again) will improve the clarity of the paper. Thank you again for the comment.
>
> 3.2: More trials are also needed to alleviate any suspicion of lucky trials.
> - We will increase the experiment trials to 10 to make the results more convincing.
>
> 3.3: Returns are defined from an initial distribution that is stationary for the policy. While this makes sense in well-mixing domains, the experiment domains are not well-mixing for most policies during training. The definition of beta(a|s) as a mixture of past actors is inconsistent with the sampling method.
> - Thank you for pointing them out. We will correct them.
>
> 3.4: In the first paragraph of Section 3.3: "[...] the quality of a guide actor mostly depends on the accuracy of Taylor's approximation." What else does it depend on? Then: "[...] the action a_0 should be in a local vicinity of a."; and "[...] the action a_0 should be similar to actions sampled from pi_theta(a|s)." What do you mean "should"? In order for the Taylor approximation to be good?
> - Beside the accuracy of Taylor’s approximation, the guide actor is determined by the step-size parameters eta and omega which depend on the accuracy of sample averages for the dual function hat{g}. The sample size can be large since we can use off-policy samples. The sample size of 256 provided a good trade-off between performance and computation time in our experiments. Regarding the latter two sentences about “a”, it is correct that we require a_0 to be close to “a” in order to obtain a good Taylor’s approximation. We will make these sentences clearer in the revise version.
>
> 3.5: Why/when is the 2nd term in (19) small?
> - The second term is inverse proportion to exp(Q(s,a)) and is small for high values of Q(s,a). It also vanishes when we compute its expectation over a softmax policy pi(a|s) = exp(Q(s,a))/Z with a normalizer Z. However, this is not the case in our setting since the guide actor does not converge to such a softmax policy unless eta -> 0 and omega -> 1. We will consider alternative Hessian approximations such as BFGS updates in future work.
>
> 3.6: The parametrization nu of \hat{Q} is never specified in Section 3.6. This is important in order to evaluate the complexities involved in computing its Hessian.
> - For a Gauss-Newton approximation, the computation cost is determined by that of gradient computation and an outer-product operation. The cost of outer-product is low. The cost of gradient computation depends on the parameterization of hat{Q}. The cost is inexpensive for simple models such as a linear model: hat{Q} = nu’*phi(s,a). For neural network models, the gradients are computed by automatic-differentiation and its cost depends on network architecture. We will include this discussion in Section 3.6.

---

### Author Response · Authors · 2017-12-11
**Author response**

We thank all the reviewers for the constructive reviews.

To explain our motivation more clearly, we will include a subsection titled “Second-order Methods for Policy Learning” under the Background section in the revise paper. Its purpose is 1) to discuss existing second-order methods on deep RL and their computational issue, and 2) to motivate the use of a guide-actor to avoid the issue. These are briefly explained in our responses to 1.1 and 1.2 to the review 1. The revised paper will be submitted as soon as possible. Below, we address each review in the comment.

---

> ### Author Response · Authors · 2018-01-05
> **Changes in the revision**
>
> Dear reviewers,
>
> We have revised the paper according to the comments and suggestions. The following changes are made to the paper:
> 1. We include a new paragraph in Section 1 and a new subsection in Section 2 to explain our motivation more clearly.
> 2. We include a discussion about the computation time at the end of Section 5.
> 3. We improve the explanation about Taylor's approximation in the first paragraph of Section 3.3 and about the Hessian approximation in Section 3.4.
> 4. We include a discussion about the parameterization of the Q-function at the end of Section 3.6.
> 5. The number of experimental trials is increased from 5 to 10. The number of training time steps is also increased from 700,000 to 1,000,000.
> 6. Grammatical errors have been corrected.

---

### Decision · Program_Chairs · 2018-01-29
**ICLR 2018 Conference Acceptance Decision**

**Decision:**

Accept (Poster)

**Comment:**

The reviewers agree that the formulation is novel and interesting, but they raised concerns regarding the motivation and the complexity of the approach. I find the authors' response mostly satisfying, and I ask them to improve the paper by incorporating the comments.

Detailed comments:
The maximum-entropy objective used in Eq. (13) reminds me of maximum-entropy RL objective in previous work including [Ziebart, 2010], [Azar, 12], [Nachum, 2017], and [Haarnoja, 2017].